# A therapeutic antibody targeting osteoprotegerin attenuates severe experimental pulmonary arterial hypertension

Nadine D. Arnold [1,9], Josephine A. Pickworth [1,9], Laura E. West [1,9], Sarah Dawson[1,9], Joana A. Carvalho[2], Helen Casbolt[1], Adam T. Braithwaite [1], James Iremonger [1], Lewis Renshall[1], Volker Germaschewski[2], Matthew McCourt[2], Philip Bland-Ward[2], Hager Kowash [1], Abdul G. Hameed [1], Alexander M.K. Rothman [1], Maria G. Frid[3], A.A. Roger Thompson [1], Holly R. Evans [4], Mark Southwood[5], Nicholas W. Morrell [5], David C. Crossman [6], Moira K.B. Whyte [7], Kurt R. Stenmark [3], Christopher M. Newman[1], David G. Kiely[1,8], Sheila E. Francis [1] & Allan Lawrie [1]*

Pulmonary arterial hypertension (PAH) is a rare but fatal disease. Current treatments increase life expectancy but have limited impact on the progressive pulmonary vascular remodelling that drives PAH. Osteoprotegerin (OPG) is increased within serum and lesions of patients with idiopathic PAH and is a mitogen and migratory stimulus for pulmonary artery smooth muscle cells (PASMCs). Here, we report that the pro-proliferative and migratory phenotype in PASMCs stimulated with OPG is mediated via the Fas receptor and that treatment with a human antibody targeting OPG can attenuate pulmonary vascular remodelling associated with PAH in multiple rodent models of early and late treatment. We also demonstrate that the therapeutic efficacy of the anti-OPG antibody approach in the presence of standard of care vasodilator therapy is mediated by a reduction in pulmonary vascular remodelling. Targeting OPG with a therapeutic antibody is a potential treatment strategy in PAH.

[1] Department of Infection, Immunity and Cardiovascular Disease, University of Sheffield, Sheffield S10 2RX, UK. [2] Kymab Ltd, Babraham Research Campus, Cambridge CB22 3AT, UK. [3] Cardiovascular Pulmonary Research Laboratories, Departments of Pediatrics and Medicine, University of Colorado Anschutz Medical Campus, Aurora, CO 80045, USA. [4] Department of Chemistry, University of Sheffield, Sheffield S3 7HF, UK. [5] Department of Medicine, University of Cambridge School of Clinical Medicine, Addenbrooke's and Papworth Hospital, Cambridge CB2 0QQ, UK. [6] School of Medicine, University of St. Andrews, St, Andrews KY16 9AJ, UK. [7] MRC/University of Edinburgh Centre for Inflammation Research, University of Edinburgh, The Queens Medical Research Institute, Edinburgh EH16 4TJ, UK. [8] Sheffield Pulmonary Vascular Disease Unit, Sheffield Teaching Hospitals Foundation Trust, Royal Hallamshire Hospital, Sheffield S10 2JF, UK. [9] These authors contributed equally: Nadine D. Arnold, Josephine A. Pickworth, Laura E. West, Sarah Dawson. *email: a.lawrie@sheffield.ac.uk

Pulmonary arterial hypertension (PAH) is a devastating disease driven by a sustained pulmonary-specific vasoconstriction which triggers a progressive pulmonary vasculopathy that leads to right heart failure[1]. Early endothelial cell dysfunction is thought to be an initiating event in the development of PAH. The subsequent proliferation of multiple resident cell types including pulmonary artery smooth muscle cells (PASMC), endothelial cells (PAEC) and fibroblasts is critical to the vascular remodelling. The infiltration of circulating inflammatory and mesenchymal cells has been shownt to play an important role in regulating disease pathogenesis[2–5]. Current therapies for PAH are effective in relieving symptoms and improve survival[6]; however, their effects are often transient and importantly do not stop the progressive pathological changes[7]. PAH remains an orphan disease with no cure other than transplantation.

The molecular and cellular mechanisms involved in the pathogenesis of PAH are complex and involve cross-talk between several signalling pathways including the transforming growth factor beta (TGF-β)/bone morphogenetic protein (BMP) axis[8], growth factors (e.g. PDGF)[9] and vasoactive proteins (e.g. vasoactive intestinal peptide (VIP)[10] and endothelin-1 (ET-1)[11] (reviewed with respect to anti-remodelling therapies in ref.[5]). We previously reported that tumour necrosis factor (TNF) related apoptosis inducing-ligand (TRAIL) is also a critical mediator of PAH in experimental models[12]. We[13,14] and others[15] have reported that osteoprotegerin (OPG, Tnfrsf11b), a secreted glycoprotein belonging to the TNF receptor superfamily capable of binding to TRAIL, is elevated in the lungs and sera from patients with idiopathic PAH (IPAH). OPG is a potent mitogen and migratory stimulus of PASMCs in vitro[13]. Jia et al. have demonstrated that mice lacking OPG display an attenuated PAH phenotype in the Sugen5416 plus hypoxia (SuHx) model[15].

We report here that OPG expression is elevated in the mouse SuHx model, and in a different strain of OPG$^{-/-}$ mice, the PAH phenotype is similarly attenuated (Supplementary Figure 1). Levels of OPG also increase consequently with PAH development in the monocrotaline (Mct) rat (Supplementary Figure 2). Furthermore, we demonstrate in vitro that OPG binds to Fas receptor to activate cell proliferation, migration and survival pathways. Finally, using a human OPG antibody we demonstrate a robust therapeutic effect on established and severe PAH. Importantly, the efficacy of our approach was mediated through both improved haemodynamics and pulmonary vascular remodelling. The haemodynamic efficacy of our approach was at least equivalent to current standard of care PAH therapies (used in 10–50-fold excess in these rat models). Combination of current PAH therapies with our anti-OPG antibody demonstrated an improved response in both haemodynamics and pulmonary vascular remodelling over standard of care PAH therapies alone.

## Results

**OPG antibody treatment reverses PAH in HFD-ApoE$^{-/-}$ mice.** Studies by Jia et al[15], and confirmed by us, demonstrate the requirement for OPG expression to develop the full PAH phenotype in the mouse SuHx model (Supplementary Figure 1), we also demonstrated the increase of OPG expression with development of PAH in the monocrotaline rat model (Supplementary Figure 2). We next sought to determine whether OPG was a tractable therapeutic target in PAH models of established disease. We investigated the effect of genetic deletion of OPG in the Paigen high fat, high cholate containing diet (HFD) fed ApoE$^{-/-}$ mouse as a model with severe and progressive (non-resolving) pulmonary vascular remodelling[12,16]. Despite previous reports from other groups[17–19], we were unable to successfully breed and

maintain mice double deficient for ApoE and opg. We subsequently generated heterozygous ApoE mice (ApoE$^{+/-}$), and mice heterozygous for ApoE but homozygous deficient for OPG (ApoE$^{+/-}$/OPG$^{-/-}$). ApoE$^{+/-}$ mice developed PAH in response to 8 weeks of feeding HFD, consistent with our previously published data[12,16]. HFD-fed ApoE$^{+/-}$/OPG$^{-/-}$ were protected from developing increased RVSP (Fig. 1a) with no significant difference in left ventricular end-systolic (LVESP) or end-diastolic (LVEDP) pressure, in either strain (Fig. 1b–c). There was no statistically significant difference in cardiac index (CI) between HFD-fed ApoE$^{+/-}$ and ApoE$^{+/-}$/OPG$^{-/-}$ (Fig. 1d). Analysis of pulmonary vascular remodelling confirmed that the reduced RVSP in the ApoE$^{+/-}$/OPG$^{-/-}$ was associated with a significantly lower media/CSA of small pulmonary arteries (Fig. 1e–f). We next examined whether treatment of established PAH with a polyclonal anti-OPG antibody could stabilise or induce disease regression in the HFD-ApoE$^{-/-}$ model. In a separate group of animals, phenotype was confirmed after 8 weeks feeding of ApoE$^{-/-}$ mice with HFD. The remaining mice were then randomly assigned to receive blinded treatment with either a polyclonal anti-OPG antibody or IgG control for 4 weeks (Fig. 1g). Compared to HFD-fed ApoE$^{-/-}$ mice phenotyped after 8 weeks, the mice treated with the IgG control antibody displayed an increase in disease severity (Fig. 1h–i). In contrast, mice treated with the anti-OPG antibody demonstrated a significant increase in pulmonary artery acceleration time (PAAT) (Fig. 1h) and reduction of RVSP (Fig. 1i). There was no significant effect of disease, or OPG antibody treatment on LVESP (Fig. 1j). The beneficial haemodynamic response achieved by anti-OPG antibody treatment was associated with a reduction in media/CSA (Fig. 1k–l) that was associated with fewer proliferating and more apoptotic cells (Fig. 1l). Since OPG is linked with bone remodelling[20] we examined whether antibody blockade of OPG would induce an osteoporotic phenotype but no detrimental effect of the anti-OPG antibody treatment was observed on either bone volume, trabecular number or trabecular thickness as assessed by microCT analysis (Fig. 1m–o).

**Bone marrow-derived OPG drives PAH in the murine SuHx model.** To determine if the source of OPG responsible for driving disease was originating from tissue resident, or bone marrow-derived cells we next examined the disease phenotype in chimeric mice generated by bone marrow transplantation (BMT). Mice lacking tissue OPG displayed significantly reduced serum levels of OPG (Fig. 2a) but were not protected from developing PAH (Fig. 2b–f). In contrast, mice lacking OPG in bone marrow only (red dots, Fig. 2), were protected from developing PAH (Fig. 2b–f). The presence of OPG was noted within remodelled pulmonary arteries from mice that developed PAH (Fig. 2g) suggesting OPG expressing cells might be recruited from a bone marrow source.

We next sought to investigate candidate bone marrow-derived cell-types that could release OPG and drive the PAH pathophysiology. Since both endothelial[21] and mesenchymal[22] progenitors have been implicated in PAH, and are present in remodelled arteries we investigated the expression of OPG in PASMCs (SMC), pulmonary artery fibroblasts (PA-Fib) and fibrocytes isolated from the hypoxic neonatal calf model of PAH[23] and blood outgrowth endothelial cells (BOEC)[24]. OPG expression was 2-fold higher in both PA-Fibs and SMCs, but dramatically higher in fibrocytes isolated from hypoxic calves with PAH compared to controls (Fig. 2h). We subsequently performed immunohistochemical analysis of the remodelled pulmonary arterioles from the hypoxic neonatal calf model and observed a marked increase in diffuse OPG staining throughout the lesions and in the number

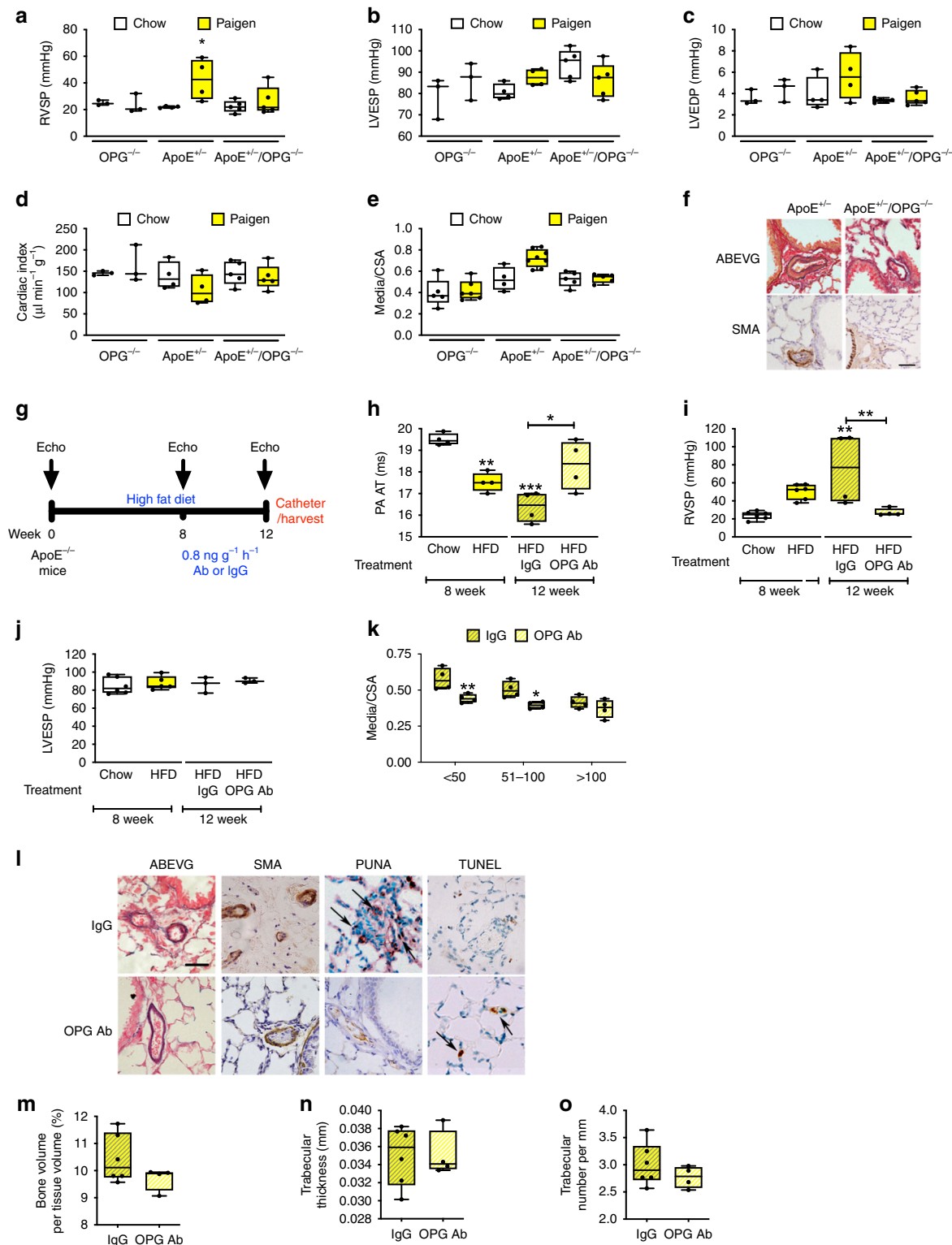

of OPG positive cells within the vessel wall, particularly in the adventitial outward remodelled parts of the artery (Fig. 2i). In BOECs, whereas vascular endothelial growth factor (VEGF) enhanced proliferation of BOEC obtained from healthy and IPAH donors, OPG only induced proliferation in BOECs derived from IPAH patients (Fig. 2j). Since OPG is naturally secreted we postulated from these data that BM-derived cells may be secreting and in turn responding to OPG alongside resident PASMCs to drive pulmonary vascular remodelling.

**OPG regulates genes important in PH/PAH pathogenesis**. To gain mechanistic insight into how OPG might regulate the pro-proliferative PASMC phenotype, we examined the transcriptome and intracellular signalling mediated by OPG in human PASMCs. Microarray analysis of PASMC mRNA identified 1900 probes from the microarray that were significantly regulated by OPG. Utilising the full transcriptomic analysis we performed pathway analysis using Signalling Pathway Impact Analysis (SPIA)[25] and identified 13 KEGG pathways as being significantly perturbed by

**Fig. 1** Genetic deletion of OPG prevents and antibody treatment reverses PAH. Panels (**a–f**) are data obtained from high fat diet (HFD) fed $OPG^{-/-}$, $ApoE^{+/-}$, and $ApoE^{+/-}/OPG^{-/-}$ mice to determine the requirement of OPG for the development of PAH. Panels (**g–o**) are data obtained high fat diet (HFD) fed $ApoE^{-/-}$ mice treated with IgG or OPG antibody to determine if OPG antibody treatment can reverse established PAH. Bar graphs (**a, i**) show right ventricular systolic pressure (RVSP), (**b, j**) left ventricular end-systolic pressure (LVESP), (**c**) left ventricular end-diastolic pressure, (**d**) cardiac index, (e&**k**) the degree of medial wall thickness as a ratio of total vessel size (Media/CSA), (**f**) representative photomicrographs of serial lung sections stained with Alcian Blue Elastic van Gieson (ABEVG) or immunostained for α-smooth muscle actin (α-SMA). Panel (**g**) demonstrates a schema from the therapeutic intervention with polyclonal mouse OPG antibody. (**h**) pulmonary artery acceleration time (PA AT). **l** Representative photomicrographs of serial lung sections from $ApoE^{-/-}$ mice fed on Paigen diet for 12 weeks. Sections were stained with ABEVG or α-SMA, proliferating cell nuclear antigen (PCNA) or Terminal deoxynucleotidyl transferase dUTP nick end labelling (TUNEL). Bar graphs show femoral trabecular bone volume (%) (**m**), trabecular thickness (mm) (**n**), trabecular number ($mm^{-1}$) (**o**), bars represent mean with error bars showing the standard error of the mean. Box and Whisker plots represent the interquartile range (box) with the line representing the median and whisker the full range of the data, each animal is represented by a dot in each graph; panels (**a–f**) $OPG^{-/-}$ $n = 3$ per group, $ApoE^{+/-}$ $n = 4$ per group, $ApoE^{+/-}/OPG^{-/-}$ $n = 5$ per group. * $p < 0.05$, ** $p < 0.01$, *** $p < 0.001$ compared to $OPG^{-/-}$ or chow-fed mice following a two-way ANOVA followed by Bonferroni's multiple comparisons test, or were only two groups, unpaired t-tests. All images are presented at their original magnification ×400, scale bar represent 50 μm

stimulation with OPG, most notably TGFβ signalling, cytoskeletal organisation, motility and survival pathways (Fig. 3a). To filter the data we first applied gene enrichment utilising a previously curated PAH gene list[26]. This highlighted 57 genes either previously associated with PH/PAH, or in key cellular mechanisms important in disease pathogenesis (Fig. 3b). Further analysis of a selection of these differentially regulated genes with TaqMan PCR validated several genes previously described as important in the pathogenesis of PH/PAH, specifically *TRAIL, PDGFRA, tenascin-C, VEGFA,* and *caveolin-1,* as all being significantly up-regulated by OPG, and the *VIP* receptor as being significantly down-regulated by OPG (Fig. 3c). To examine the intracellular signalling pathways we performed a Kinex™ antibody microarray (KAM) and identified 63 from 800 phosphorylation and pan-specific antibodies that were significantly regulated by OPG at either 10, 60 min, or both (Supplementary Figure 3). Significantly regulated proteins included a number of pro-survival, anti-apoptotic and cell cycle (Fig. 3d) proteins and members of the NF-5β pathway (Fig. 3e). Several proteins were validated by western immunoblotting, further emphasising activation of MAPK signalling (pERK1/2), anti-apoptotic proteins (pHsp27, CDK5) and mammalian target of rapamycin (mTOR) and cell cycle (CDK4) (Fig. 3f).

**OPG binds to Fas receptor on PASMCs**. Given the effect of OPG on cell phenotype and the intracellular signalling identified, we felt that OPG may be acting as a ligand and signalling through a previously undescribed receptor. To identify the signalling receptor for OPG, we conducted a reverse transfection membrane protein array (Retrogenix, UK). Primary and secondary screens identified six twice-validated OPG-protein interacting partners, RANKL (*tnfsf11*), syndecan-1 (SDC-1), Fas, IL1-receptor accessory protein (IL-1RAcP), growth associated protein 43 (GAP43) and TMPRSS11D (Fig. 4a). We have previously reported that levels of SDC-1 were undetectable, and RANKL was only detected at low level in IPAH tissues[13]. Therefore we focused on investigating the four OPG-interacting proteins identified (Fas, IL-1RAcP, Gap43 and TMPRSS11D). Expression of *TMPRSS11D* was undetectable in mRNA isolated from PASMCs. The RNA expression of *Fas, IL-RAcP* and *GAP43* was confirmed in PASMCs, with *Fas* being the most abundantly expressed, and further induced by OPG (Fig. 4b). Similarly, *Fas* mRNA was more highly expressed in PASMCs from patients with IPAH compared to healthy controls (Fig. 4c). Since Fas was the most abundantly expressed putative receptor we performed immunoprecipitation on lysates from PASMCs stimulated with OPG to validate binding. In both PASMC lysates and recombinant protein preparations, immunoprecipitation with a Fas monoclonal antibody pulled down a 50 kDa band that stained positive following anti-

OPG immunoblotting (Fig. 4d). Furthermore, Fas immunoreactivity strongly associated with both remodelled pulmonary arteries, and the right ventricle of patients with IPAH (Fig. 4e) compared to controls. Investigation of rat lung isolated from control (saline) and moncrotaline rats, as well as control (normoxic) and SuHx rats also demonstrate a significant increase in expression of both Fas gene expression (Fig. 4f) and protein expression within remodelled pulmonary arterioles (Fig. 4g).

**Fas regulates OPG signalling and phenotype in PASMCs**. To determine the functional and signalling consequences of the OPG-Fas interaction, PASMCs were stimulated with OPG after pre-incubation with an anti-human Fas neutralising antibody. Blockade of the Fas receptor prevented OPG induction of *PDGFRA, TNC, VEGFA* and *CAV1* gene expression (Fig. 5a–d) but interestingly not *TRAIL* (Fig. 5e). To validate the functional role of the OPG-Fas interaction, we used the well-described model of FasL/TRAIL-induced apoptosis of HT1080 cells[27]. Pre-incubation of HT1080 cells with OPG significantly blocked both TRAIL but also FasL-induced apoptosis, as measured by Caspase3/7 activation (Fig. 5f) indicating that OPG can antagonise FasL–Fas binding. To further examine this in a disease-relevant cell type, we examined the effect of Fas neutralisation on OPG stimulated human PASMC. Fas neutralisation significantly reduced OPG-induced transwell PASMC migration (Fig. 5g) and suppressed OPG-induced proliferation (Fig. 5h). However, Fas neutralisation had no effect on PDGF-induced proliferation (Fig. 5h). The observed increase in TRAIL expression following ligation of Fas receptor with either the Fas neutralising antibody, or OPG itself (Fig. 5e), led us to hypothesise that the remaining proliferation in response to OPG where Fas is neutralised may be mediated by TRAIL (since we have previously described TRAIL as a PASMC mitogen[12]). Pre-incubation with both an anti-TRAIL antibody and anti-Fas antibody significantly reduced OPG-induced PASMC proliferation to near baseline levels (Fig. 5h) suggesting a direct activation of TRAIL-induced proliferation in PASMCs following Fas binding. Based on these and earlier data (Fig. 3), we therefore propose that OPG binding to Fas causes intracellular kinase signalling, including phosphorylation of ERK1/2, CDK4/5 leading to the activation of multiple genes associated with PAH, notably TRAIL. This induces a pro-survival, migratory and proliferative phenotype promoting pulmonary vascular remodelling and PAH (Fig. 5i). Furthermore, we propose that inhibition of OPG, e.g. via antibody blockade, will prevent this signalling and subsequent alteration in pro-PAH gene expression leading to a reversal of pulmonary vascular remodelling, normalisation of pulmonary vascular resistance and inhibition of PAH via alteration in the proliferation, migration and apoptosis of pulmonary vascular cells (Fig. 5j).

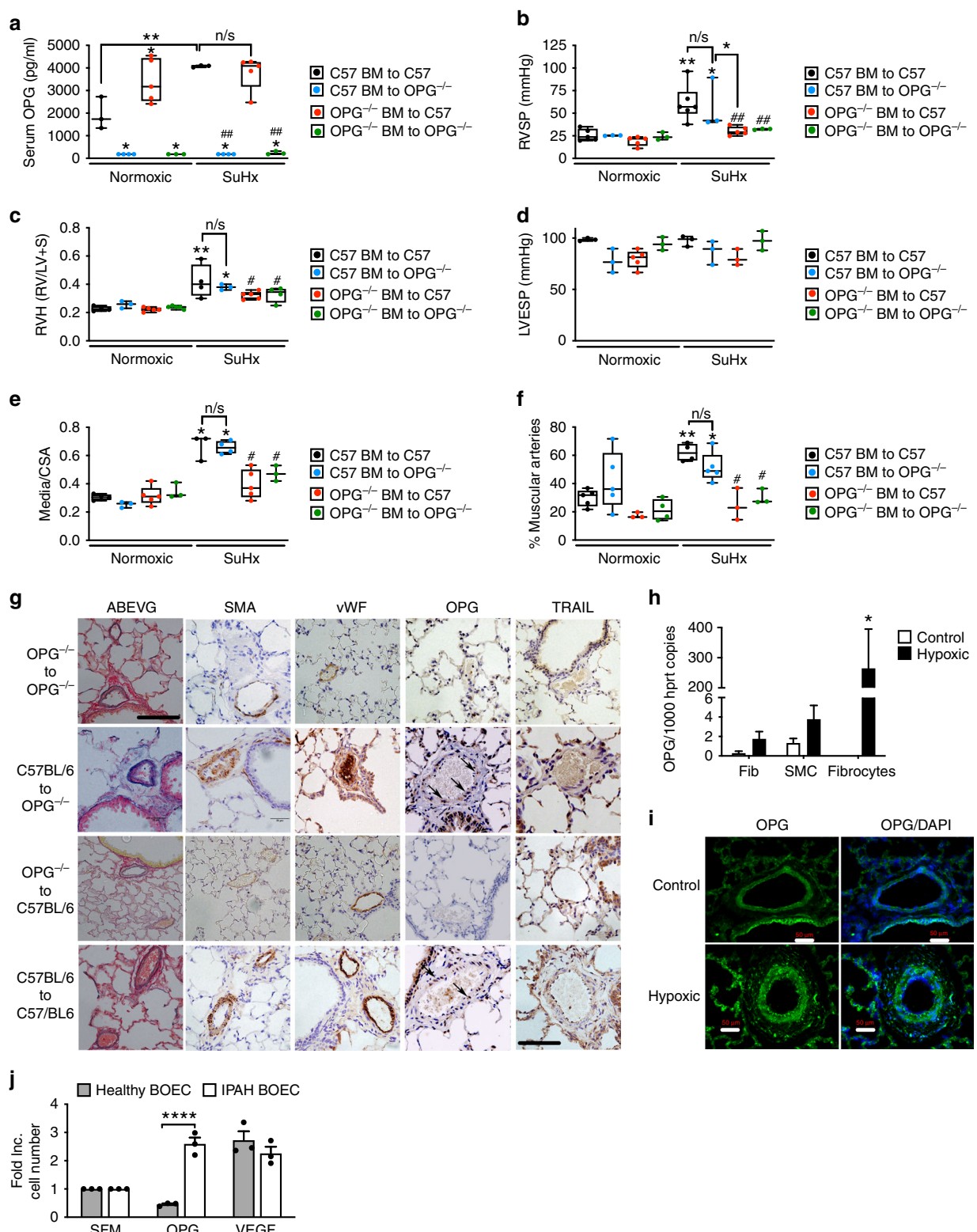

**Identification of a lead therapeutic anti-OPG antibody**. Our data indicate that OPG is a likely therapeutic candidate for PAH. Using the KyMouse™ system[28] we generated a diverse panel of high affinity anti-human OPG monoclonal antibodies with cross-reactivity to rat and cynomolgus monkey, displaying distinct neutralisation profiles and varying ability to block the interaction of OPG with TRAIL and RANKL (Supplementary Figure 4). Selected antibodies were chosen to cover a spectrum of partial and full inhibition of OPG-TRAIL and OPG-RANKL binding. OPG-FAS signalling was examined later (see below). Four candidate anti-OPG antibodies (Supplementary Figure 4e) were tested for their ability to attenuate the development of monocrotaline-induced PAH (Fig. 6a). Weekly delivery of 3 mg kg$^{-1}$ antibody or IgG control resulted in the expected levels of circulating plasma antibody (Fig. 6b). Analysis of the complete dataset identified the Ky3 antibody as having a significant

**Fig. 2** Bone marrow cell derived OPG is required to initiate PAH in the mouse SuHx model. Bar graphs show (**a**) serum levels of OPG, (**b**) right ventricular systolic pressure (RVSP), (**c**) right ventricular hypertrophy (RVH), (d) left ventricular end-systolic pressure (LVESP), (**e**) the degree of medial wall thickness as a ratio of total vessel size (Media/CSA) in small pulmonary arteries pulmonary arteries less than 50 µm, (**f**) the relative percentage of muscularised pulmonary arteries less than 50 µm (<50 µm) in diameter. Representative photomicrographs (**g**) of serial lung sections from bone marrow-transplanted (BMT) mice. Sections were stained with Alcian Blue Elastic van Gieson (ABEVG), or immunostained for α-smooth muscle actin (α-SMA), von Willebrand factor (vWF), OPG, or TRAIL. Panel (**h**) shows OPG gene expression from RNA-seq performed on control and PAH-derived pulmonary artery smooth muscle cells (SMC), pulmonary artery fibroblasts (Fib) and fibrocytes obtained from the hypoxic neonatal calf model of PAH. Representative photomicrographs of lung sections from the hypoxic neonatal calf stained with OPG (**i**). Proliferation of blood outgrowth endothelial cells (BOEC) from patients with IPAH and healthy controls (**j**). Box and Whisker plots represent the interquartile range (box) with the line representing the median and whisker the full range of the data, each animal is represented by a dot in each graph. C57-C57 BMT $n = 6$ for each group, OPG$^{-/-}$-OPG$^{-/-}$ $n = 3$ for each group, C57-OPG$^{-/-}$ $n = 3$ and OPG$^{-/-}$-C57 $n = 5$ for each group. * $p < 0.05$, ** $p < 0.01$,*** $p < 0.001$ compared to C57-C57 BMT Normoxic mice unless otherwise stated, # $p < 0.05$, ## $p < 0.01$ compared to C57–C57 SuHx mice following one-way ANOVA with Bonferroni's multiple comparisons post hoc test. All images are presented at their original magnification x400, scale bars represent 50 µm

attenuation on markers of PAH including RVSP (Fig. 6c), RVH (Fig. 6d) and ePVRi (Fig. 6e). There was no significant effect of either treatment on LVESP (Fig. 6f). Immunohistochemical analysis of the lung demonstrated a significant reduction in the media/CSA area (Fig. 6g) and percentage of thickened sub-50 µm pulmonary arterioles (Fig. 6h–i) in rats treated with the Ky3 antibody. Interestingly rats treated with either the commercial polyclonal anti-mouse OPG antibody (AF459), which demonstrated partial efficacy in the SuHx and efficacy in the ApoE$^{-/-}$ mouse (Fig. 1), or Ky3 resulted in a significant increase in serum levels of OPG (Fig. 6j), possibly due to retention of antibody bound OPG in the circulation rather than allowing it to access the vessel wall.

**Ky3 inhibits OPG-induced phenotype and NF-κβ activation.** Once Ky3 was identified as the lead candidate antibody for further development we confirmed that Ky3 inhibited OPG-induced proliferation (Fig. 7a) and migration (Fig. 7b) in human PASMC in vitro. Having previously identified an NF-κβ response to OPG (Fig. 3e) we investigated, and show that Ky3 inhibits this activation (Fig. 7c).

**Ky3 attenuates severe PAH.** Antibody Ky3 was tested therapeutically in two rat models with severe established PAH, Mct and SuHx. Rats were exposed to Sugen5416 and hypobaric hypoxia (18,000 ft, equivalent to 10.8% O$_2$) for 3 weeks before returning to room air for 3 weeks to allow the progression of pulmonary vascular remodelling. Rats were then randomised into groups to receive either sildenafil (50 mg kg$^{-1}$ per day), Ky3 (3 mg kg$^{-1}$ per week) or IgG (3 mg kg$^{-1}$ per week) control antibody from week 6 for 3 weeks (Fig. 8a). Sustained levels of Ky3 and IgG were maintained throughout the study (Fig. 8b). PA AT decreased from week 0 to week 6 as disease progressed. There was a trend for increased PA AT in sildenafil vs SuHx and Ky3 vs IgG4 treated animals but this did not reach significance (Fig. 8c). Sildenafil treated rats showed an increase in cardiac output (CO) (Fig. 8d). Treatment with sildenafil and Ky3 significantly reduced RVSP (Fig. 8e) compared to untreated and IgG4 controls, respectively. RV arterial elastance (RV Ea) and ePVRi were significantly reduced only by Ky3 (Fig. 8f–g), treatment with sildenafil and Ky 3 significantly reduced RVH (Fig. 8h). There was no significant effect of any treatment on LVESP (Fig. 8i) indicating specific effects on the pulmonary circulation. Immunohistochemical analysis of the lung demonstrated that the haemodynamic changes induced by anti-OPG treatment were associated with a reduction in both the media/CSA (Fig. 8j) and percentage of muscularised pulmonary arterioles sub-50 µm in diameter (Fig. 8k). In contrast there was no significant effect of sildenafil on either the degree of remodelling, or the percentage of remodelled

vessels (Fig. 8j–k). To try and elucidate the different mechanisms of action of sildenafil and Ky3 we performed Caspase 3 and PCNA staining to examine the relationship between treatment and apoptosis and proliferation on serial sections within the small remodelled pulmonary arterioles (Fig. 8l). In the sildenafil and IgG4 treated groups there was evidence of apoptosis, predominantly in endothelial cells, and medial proliferation. By contrast, Ky3 treated rats appeared to have apoptosis in both endothelial and medial layers and reduced medial cell proliferation.

Plasma levels of OPG were significantly elevated in all SuHx rats compared to controls at week 6 (Fig. 8m). Consistent with previous experiments, rats treated with Ky3 displayed a significant increase in circulating OPG from week 7 through to week 9 compared to other groups (Fig. 8m). To assess any potential detrimental side-effect of anti-OPG treatment on bone turnover, microCT studies were performed on the tibia. Treatment with Ky3 had no significant effect on bone volume (Fig. 8n) or trabecular thickness (Fig. 8o) compared to IgG4 treated rats; however, there was a small but significant decrease in trabecular number (Fig. 8p) in IgG4 treated rats compared to Ky3, although Ky3 treated rats were not significantly different compared to control or SuHx rats.

In the Mct model we also observed a significant reduction in pulmonary vascular remodelling with only 2 weeks of Ky3 treatment but this did not alter the haemodynamic profile (Supplementary Figure 5). We proposed that this was due to the shorter treatment duration and, particularly advanced/severe phenotype in this instance of the model.

**Ky3 reduces tissue expression of IL-6, OPG and TRAIL.** To demonstrate that the therapeutic effects of Ky3 treatment in the SuHx rat model were associated with reduced OPG signalling, we examined the expression of OPG and identified downstream mediators in the lung tissue. Despite the increase in circulating levels of OPG (Fig. 8m), Ky3 treatment resulted in a significant reduction in OPG RNA (Fig. 9a) and protein within whole lung lysates (Fig. 9b), Similarly, levels of TRAIL were also decreased at RNA (Fig. 9c) and protein level (Fig. 9d). Treatment with Ky3 was also associated with a reduction in inflammation within the lung as shown by IL-6 RNA expression (Fig. 9e) although there was no effect on total circulating levels of IL-6 (Fig. 9f). These changes were also consistent with those observed within remodelled pulmonary arterioles by IHC (Fig. 9g).

**Ky3 and standard of care vasodilator therapy combination.** Finally, Ky3 antibody (3 mg kg$^{-1}$ per week) was then tested in comparison, and combination with, sildenafil (50 mg kg$^{-1}$

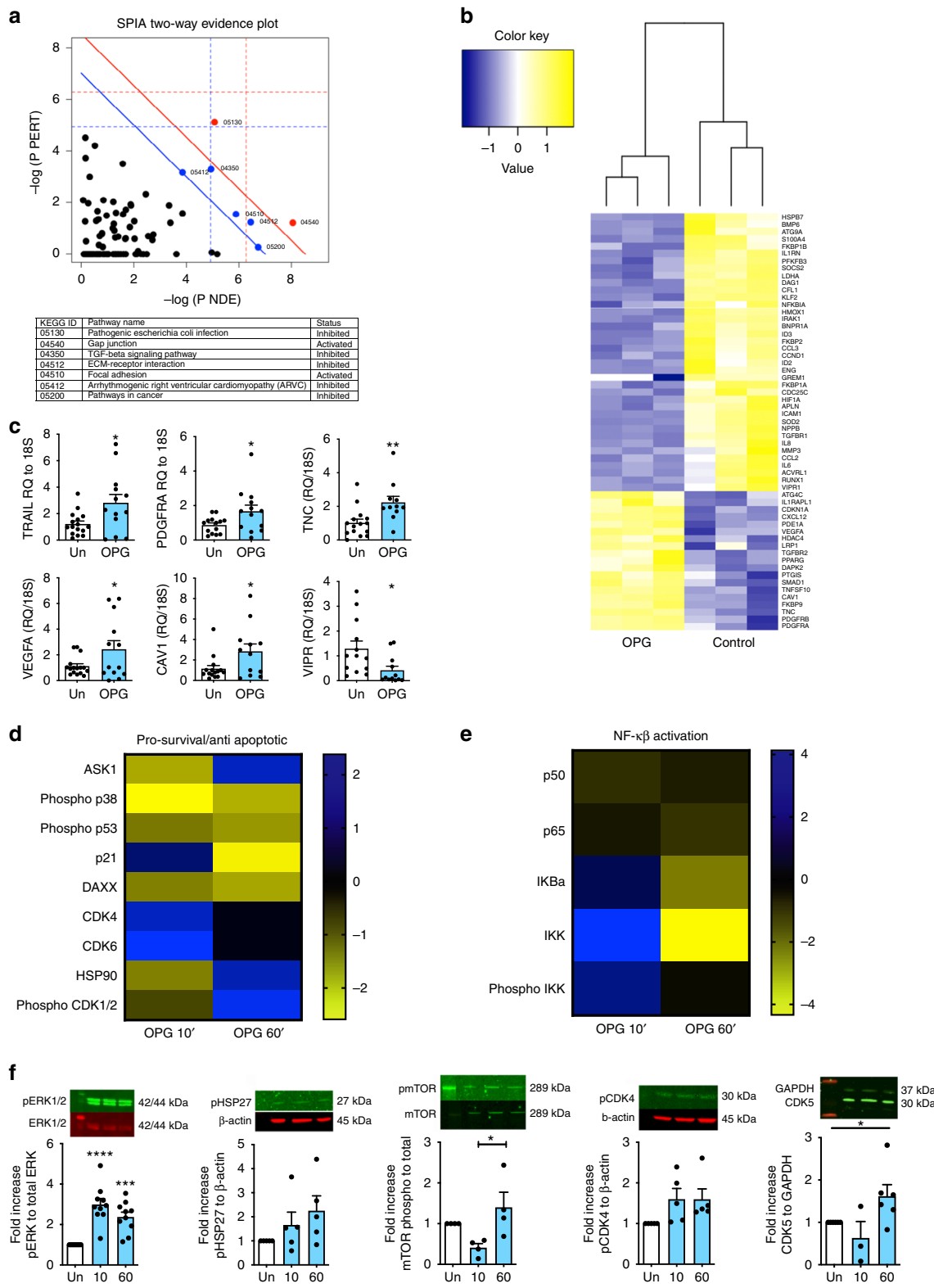

per day) or bosentan (60 mg kg−1 per day) treated rats exposed to SuHx, with IgG4 treatment as a control (Fig. 10a). There was no effect of either sildenafil or bosentan on the levels of circulating Ky3 as measured by IgG4 luminex assay (Fig. 10b). Treatment of SuHx rats with sildenafil, bosentan or Ky3 resulted in a comparable reduction PAH phenotype (Fig. 10c–h). Ky3 in combination with bosentan resulted in a significant further reduced RVSP compared to bosentan alone (Fig. 10c). Sildenafil treated

rats only demonstrated a reduction in pulmonary vascular remodelling when also receiving Ky3 (Fig. 10h). As previously demonstrated rats treated with Ky3 had increased circulating levels of OPG (Fig. 10i). Caspase 3 and PCNA staining identified an increase in apoptosis and decrease in proliferation within the small remodelled pulmonary arterioles in the lungs of rats treated with Ky3 when compared to either sildenafil or bosentan alone (Fig. 10j).

**Fig. 3** OPG activates pro-proliferative signalling and a disease-relevant transcriptome. Panel (**a**) Signalling Pathway Impact Analysis (SPIA) with each pathway represented by one dot. The pathways to the right of the red diagonal line are significant after Bonferroni correction of the global *p*-values obtained using Fisher's methods from the combination of pPERT and pNDE values, the pathways to the right of the blue line are significant after FDR correction. (**b**) shows a heat map of significant differentially regulated genes after gene enrichment against PAH-associated genes in OPG stimulated PASMCs, (**c**) TaqMan validation of gene expression microarray, TaqMan expression data normalised using $\Delta\Delta$CT with 18 s rRNA as the endogenous control gene. Panel (**d**) shows a heat map of cell cycle/CDK proteins significantly regulated by OPG at 10 and 60 min expressed as a ratio to unstimulated controls from the same time point from Kinex phospho-arrays identified, with (**e**) showing those specifically related to NF-$\kappa$β. **f** Western blot validation of Kinex array data in unstimulated (0.2% FCS, Un) or OPG-stimulated (50 ng ml$^{-1}$) PASMCs at 10 min (10) and 60 min (60) with relative band densities of phospho-ERK1/2, phospho-HSP27, phospho-mTOR, phospho-CDK4 and total CDK5 are shown by the bar graphs and representative western blot images shown above the graph. Heat maps show Z-ratio gene or protein expression. Bars represent mean with error bars showing the standard error of the mean, $n = 3$ for pooled triplicate samples (**a**, **b**), $n = 12$ (**c**), $n = 4$ (**d**, **e**), $n = 5$ (**f**) from three donors of PASMCs, dots represent experimental repeats. Bars from unstimulated cells are white, OPG stimulated blue. *$p < 0.05$, ** $p < 0.01$, *** $p < 0.001$ compared OPG-stimulated to unstimulated PASMCs using one-way ANOVA followed by Bonferroni's multiple comparisons post hoc test. When there were only two groups, unpaired t-tests were used

## Discussion

We report that OPG promotes cell survival, pro-migratory and pro-proliferative signalling in PASMCs through binding to Fas receptor. Furthermore, we demonstrate that OPG is required for full development of PAH in multiple rodent models. PAH was substantially attenuated and reversed in these models by administration of a human anti-OPG therapeutic antibody (Ky3). The mechanism for this effect was due to a reduction in pulmonary vascular remodelling indices through the modulation of proliferation and apoptosis within small pulmonary arterioles due to alterations in downstream signalling via NF-$\kappa$β, ERK, CDKs. This effect was in contrast to rats treated with sildenafil (a vasodilator and first line treatment), which displayed a similar haemodynamic response but was without effect upon pulmonary vascular remodelling. Ky3 in combination with bosentan further reduced RVSP compared to bosentan alone suggesting that anti-OPG treatment may have a benefit in addition to existing vasodilator therapy (even when used in relative excess in these rodent models compared to human use). Although new drugs[29,30] have recently been added to the treatment options available to PAH physicians, these therapies continue to target sustained pulmonary vasoconstriction. While this is a common pathophysiological feature of all forms of PAH, there is little evidence that drugs targeting the endothelin, nitric oxide or prostacyclin pathways[31] have a direct or lasting effect on pulmonary vascular cell proliferation. Indeed, they do not reverse the proliferative changes observed in PAH, emphasising the need for anti-proliferative therapies[9]. Our data demonstrate an unequivocal role for OPG in the pathogenesis of PAH via the modulation of proliferative and apoptotic changes observed in PAH. OPG has also been shown to block TRAIL binding to its receptors, a key regulator of apoptosis in sensitive cells[32], immunoregulation and immune surveillance[33,34] and in both neutrophil[35,36] and macrophage[37,38] clearance in the lung. Of particular relevance, we have previously described an important role for TRAIL in PAH[12,39] and have described how both TRAIL and OPG can be separately regulated by a number of pathways associated with PAH including BMPs, 5-HT and inflammatory cytokines[12,13].

Previous data suggest that the predominant function of OPG is to regulate osteoclastogenesis, with data from mice demonstrating that reduced OPG expression results in osteoporosis[40] and overexpression of OPG causes osteopetrosis[41] via binding to RANKL. These data perhaps suggest that therapeutic strategies targeting OPG might have detrimental effects on bone remodelling; however, encouragingly in our studies we demonstrate a positive therapeutic effect on pulmonary vascular remodelling with no significant effect on bone phenotype. OPG also binds proteins other than RANKL and TRAIL, e.g. syndecan-1, glycosaminoglycans (GAGs), von Willebrand factor and factor VIII-von Willebrand factor complex[42]. We therefore performed an

unbiased screen of around 60% of known transmembrane proteins and identified OPG binding to RANKL, syndecan-1 but also with Fas, IL-1RAcP, GAP43 and TMPRSS11D (Fig. 4). Having previously examined RANKL and syndecan-1[13], we assessed the expression of the other binders within PASMCs, and subsequently focused on Fas due its relatively high expression levels within diseased tissue and its close relationship to OPG and TRAIL (all belong to the TNF superfamily). Our data suggest that neutralisation of Fas, either by anti-Fas antibody or binding to OPG, up-regulates TRAIL expression. This may reflect a redundancy mechanism between the two death-receptor signalling pathways. FasL has been reported to induce PASMC apoptosis[43], so our data highlight another potential mechanism by which increased OPG (via Fas) may drive PAH pathology. Inhibiting FasL/Fas binding with endogenous OPG may limit the ability of FasL to cause apoptosis[44]. Indeed, we clearly show that OPG induces a pro-survival/anti-apoptotic phenotype, and activates many genes previously associated with PAH, including TRAIL, suggesting a pivotal role in the disease process. The implication that OPG can regulate the local expression of TRAIL within the vessel wall fits with our reports demonstrating that TRAIL[39], and specifically tissue-derived TRAIL[12] is required for mice to develop PAH. Of note, TRAIL was also recently described to be an important member of an immune cluster of circulating proteins that defined poor prognosis in patients with mixed aetiology PAH[45]. The relationship between cell expressed, and circulating TRAIL is however complex. TRAIL is widely expressed, including by immune cells and circulating "soluble" TRAIL requires proteolytic cleavage of the C-terminal extracellular domain of the transmembrane TRAIL protein. Whether disease is mediated by locally expressed and retained TRAIL or by released circulating TRAIL remains unclear.

The wider implications of the identified interaction between OPG and IL-1RAcP have not yet been fully examined. We[16] and others[46] have previously highlighted the importance of IL-1 in the pathogenesis of PAH but the direct effect of OPG on IL-1/IL-1R1, or IL-33/ST2 to complex with IL-1RAcP remains unclear. Similarly, the binding of OPG to GAP43 and TMPRSS11D has not been further pursued at this stage due to their low expression in diseased cells. GAP43 is reported to be a neuron-specific protein[47] and TMPRSS11D (human airway trypsin-like protease, HAT) is a type-II transmembrane trypsin-like serine protease that is largely found in sputum and expressed by bronchial ciliated endothelial cells[48]. Further work is clearly required to determine the influence of OPG in other biological processes and diseases where IL-1RAcP, GAP43 and TMPRSS11D play an important role. Our study was initially limited by the lack of availability of monoclonal anti-human OPG antibodies with cross-reactivity to rat but we overcame this by generating a suite of human monoclonal antibodies that

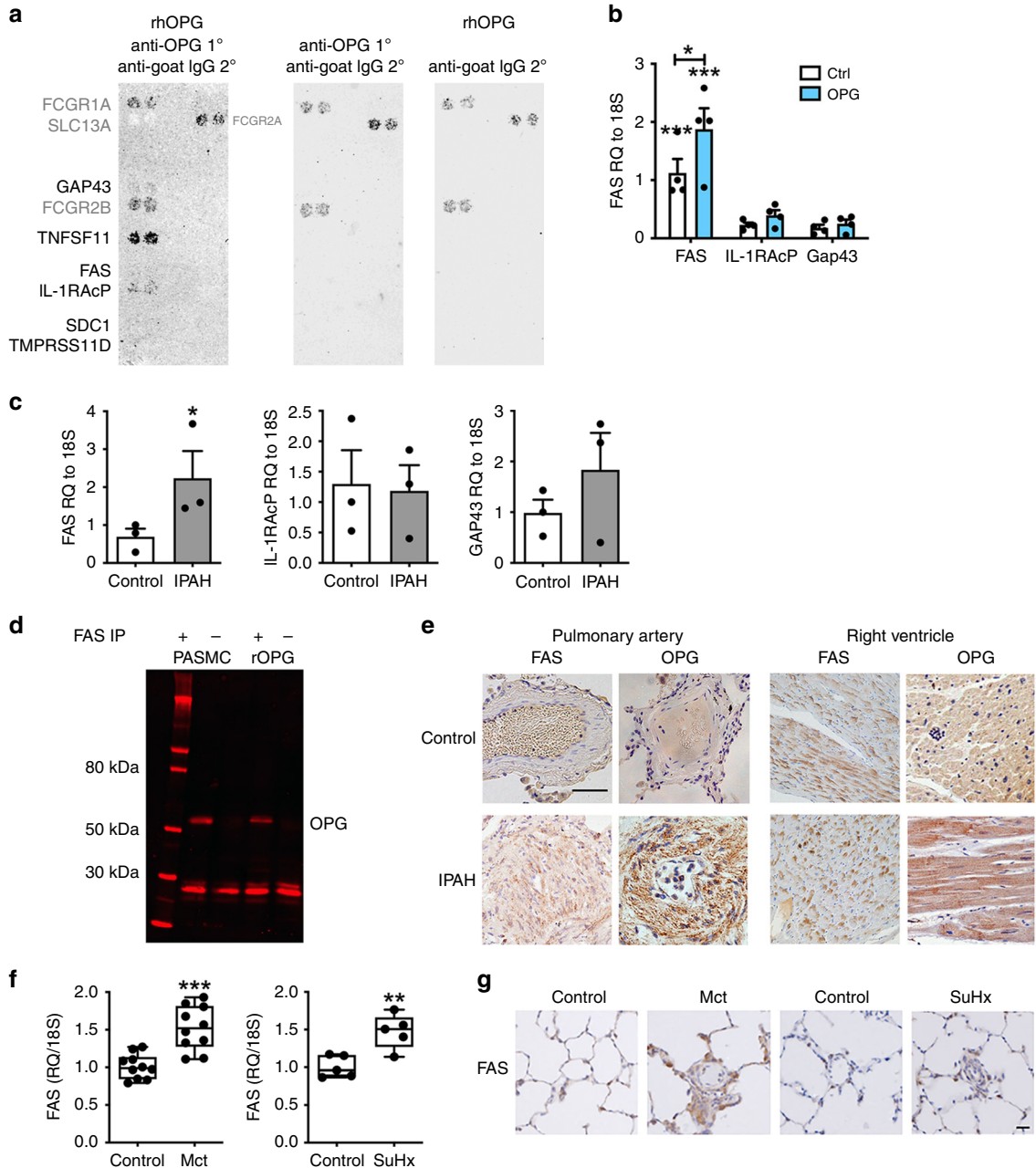

**Fig. 4** OPG binds to Fas, which is increased in IPAH lung and right ventricle. Panel (**a**) demonstrates confirmed protein binding between OPG and syndecan-1 (SDC-1), RANKL (TNFSF11), Growth Associated Protein 43 (GAP43), Fas, IL1-receptor accessory protein (IL-1RAcP) and transmembrane protease, serine 11D. **b** TaqMan expression of Fas, IL-1RAcP and GAP43 in control (white bars, 0.2% FCS) and OPG-stimulated (blue bars, 50 ng ml$^{-1}$) purchased PASMCs, and (**c**) PASMCs from patients with IPAH (grey bars) and healthy controls (white bars). **d** Anti-Fas co-immunoprecipitation of OPG in endogenous primary human PASMC lysates or recombinant protein replicated 3 times. **e** OPG and Fas are expressed within remodelled pulmonary arteries and the right ventricle of patients with IPAH. TaqMan expression of Fas in whole lung RNA (f) and protein expression in lung sections (**g**) isolated from control (saline), monocrotaline (d28), control (normoxia) and SuHx (wk9) rats. TaqMan expression data normalised using $\Delta\Delta CT$ with 18 s rRNA as the endogenous control gene. Bars represent the mean with error bars showing the standard error of the mean. Panel (**c**) $n = 4$ and panel (**d**) $n = 3$ from three individual donors, dots represent experimental repeats. * $p < 0.05$, ** $p < 0.01$, *** $p < 0.001$ following one-way ANOVA with Bonferroni's multiple comparisons post hoc test. When there were only two groups, unpaired t-tests were used. Scale bar represents 25 µm

included Ky3. Although there are limitations of each rodent model of PH used in this study, the utilisation of multiple models, each with different characteristics, combined with human data circumvent these concerns. Furthermore, the efficacy demonstrated here may not reflect the full potential effect in humans due to incomplete homology between human and rat proteins. We provide a strong body of evidence with

concordant data that OPG is a key driver in the pulmonary vascular remodelling in PAH, thereby validating it as a therapeutic target. It seems likely that Ky3 might be useful as an adjunct therapy alongside existing treatments that target vasoconstriction and we are currently exploring the potential for translation of this human therapeutic anti-OPG antibody to clinical studies in PAH.

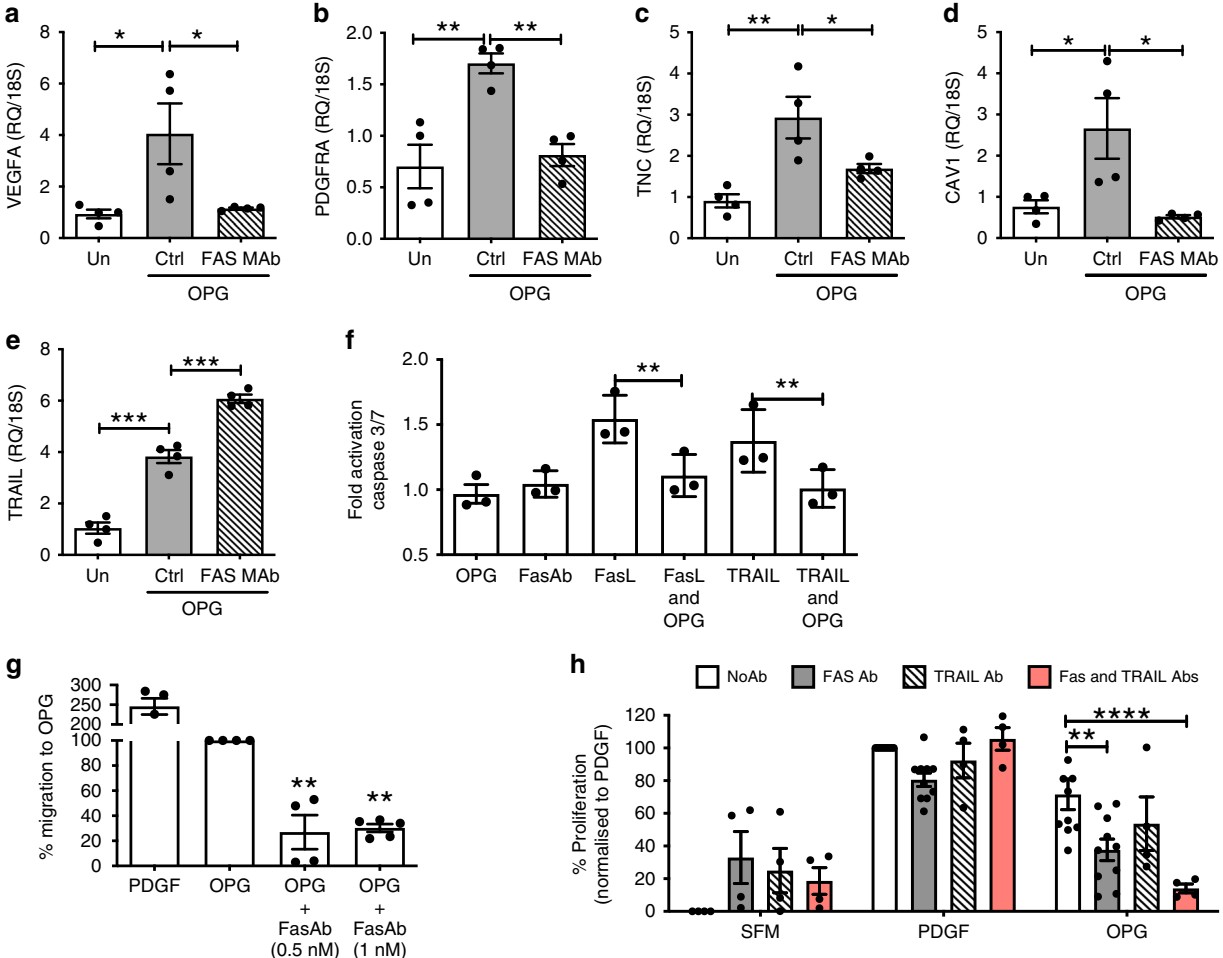

**Fig. 5** OPG-Fas interaction mediates the OPG-induced phenotypic response of PASMC. TaqMan expression of (**a**) VEGFA, (**b**) PDGFRA, (**c**) TNC, (**d**) Cav1 and (**e**) TRAIL in response to OPG in the presence (hash bars) or absence (Grey bars) of anti-Fas neutralising antibody (1500 ng ml$^{-1}$). Panel (**f**) demonstrates OPG inhibition of FasL and TRAIL-induced apoptosis in HT1080 cells. **g** PASMC migration following 6 h stimulation with PDGF (20 ng ml$^{-1}$), OPG (30 ng ml$^{-1}$) or 0.2% FCS (serum-free media, SFM), in the presence or absence of Fas neutralising antibody. **h** Proliferation of PASMCs following stimulation with OPG for 72 h in the presence or absence of Fas neutralising antibody and/or TRAIL neutralising antibody (0.5 nM). Proliferation expressed as a percentage of proliferation to PDGF. Bars represent the mean with error bars showing the standard error of the mean. Dots represent experimental repeats, Panels (**a**–**e**) ($n = 4$), panel (**f**) ($n = 3$), panel (**g**) ($n = 4$), panel (**h**) ($n = 4$ for SFM, 10 for PDGF & OPG stimulations) * $p < 0.05$, ** $p < 0.01$, *** $p < 0.001$, **** $p < 0.0001$ following one-way ANOVA with Bonferroni's multiple comparisons post hoc test

## Methods

**Animals**. All animal experiments were approved by the University of Sheffield Project Review Committee and conformed to the UK Home Office ethical guidelines. A sample size of at least four animals was used to provide greater than 95% power to detect a difference in RVSP of 10 mmHg with a SD of 3 mmHg with 95% confidence. Additional animals were studied in large group comparisons and to obtain sufficient tissue for analysis. Animals used for antibody intervention studies were randomised blindly based on weights to achieve a similar distribution of weights across all groups where possible.

Male Sprague Dawley rats were purchased from Charles River UK. PAH was induced by a single subcutaneous injection of monocrotaline (MCT, Sigma Aldrich, St. Louis, MO, USA) at 60 mg kg$^{-1}$ in rats 200–210 g, alongside saline injected control animals. For time course experiments animals were sacrificed at days 7, 14, 21 and 28. Preventative treatments with the neutralising goat polyclonal anti-OPG antibody (AF459, R&D Systems, Minneapolis, MN, USA) or control IgG isotype (AF6775, R&D systems) were administered via an Alzet 2002 mini-pump (200 µl reservoir, 0.5 µl h$^{-1}$ for 2 weeks from day 0); preventative treatments with the human monoclonal anti-OPG antibody or control IgG4 isotype were performed (3 mg kg$^{-1}$, i.p.) at day 0, 7 and 14 with the animals sacrificed at day 21. Therapeutic intervention was performed at day 21 and day 28 (3 mg kg$^{-1}$, i.p.) and animals sacrificed at day 35.

PAH was induced in male Wistar (Charles River, UK) rats of 200–220 g by a single subcutaneous injection of Sugen5416 (Tocris, Bristol, UK) at 20 mg kg$^{-1}$ followed by housing in hypobaric chambers at an equivalent of 18,000 ft for 3 weeks, followed by normobaric pressures for remaining 6 weeks. Therapeutic

treatments with the human monoclonal anti-OPG antibody or control IgG4 isotype were performed (3 mg kg$^{-1}$ per week, i.p.) alone or in combination with sildenafil (50 mg kg$^{-1}$ per day) or bosentan (60 mg kg$^{-1}$ per day) in chow from weeks 6 with animals sacrificed at week 9.

ApoE$^{-/-}$ (JAX 2052) and OPG$^{-/-}$ (JAX 010672) mice from a C57BL/6 J background were purchased from Jackson Labs. ApoE$^{+/-}$/OPG$^{-/-}$ were subsequently bred in-house. Male C57BL/6, ApoE$^{+/-}$, OPG$^{-/-}$ and ApoE$^{+/-}$/OPG$^{-/-}$ aged 10–12 week were fed normal chow (4.3% fat, 0.02% cholesterol, 0.28% sodium) or Paigen diet (18.5% fat, 0.9% cholesterol, 0.5% cholate, 0.269% sodium) for 8 weeks[8,11]. Where stated BMT was performed on male mice aged 6–8 weeks old, where each received a sub-lethal dose of whole-body irradiation (1100 rads, split into two doses, 4 h apart). Irradiated recipients then received 3–4 million cells isolated from 4 to 6 week old mice, in Hanks' balanced salt solution, by tail-vein injection[12,49]. Mice were allowed to recover for 6 weeks after bone marrow transfer prior to induction of PAH. Where stated, neutralising goat polyclonal anti-OPG antibody (AF459, R&D Systems) or control IgG isotype antibody (AF6775, R&D systems) was used. Antibodies were delivered via an Alzet 1004 micro pump (100 µl reservoir, 0.1 µl h$^{-1}$ for 4 weeks at 20 ng h$^{-1}$ (0.8 ng g$^{-1}$ h$^{-1}$). For the Sugen hypoxic model (SuHx), C57BL/6 and OPG$^{-/-}$ male mice were exposed to hypoxia (10% v/v O$_2$) for 3 weeks with weekly injections of 20 mg kg$^{-1}$ Sugen5416 (Tocris) during exposure to hypoxia[50].

**Human antibody generation**. KyMouse™ system of genetically engineered mice containing a large number of human immunoglobulin genes[28] was used for the

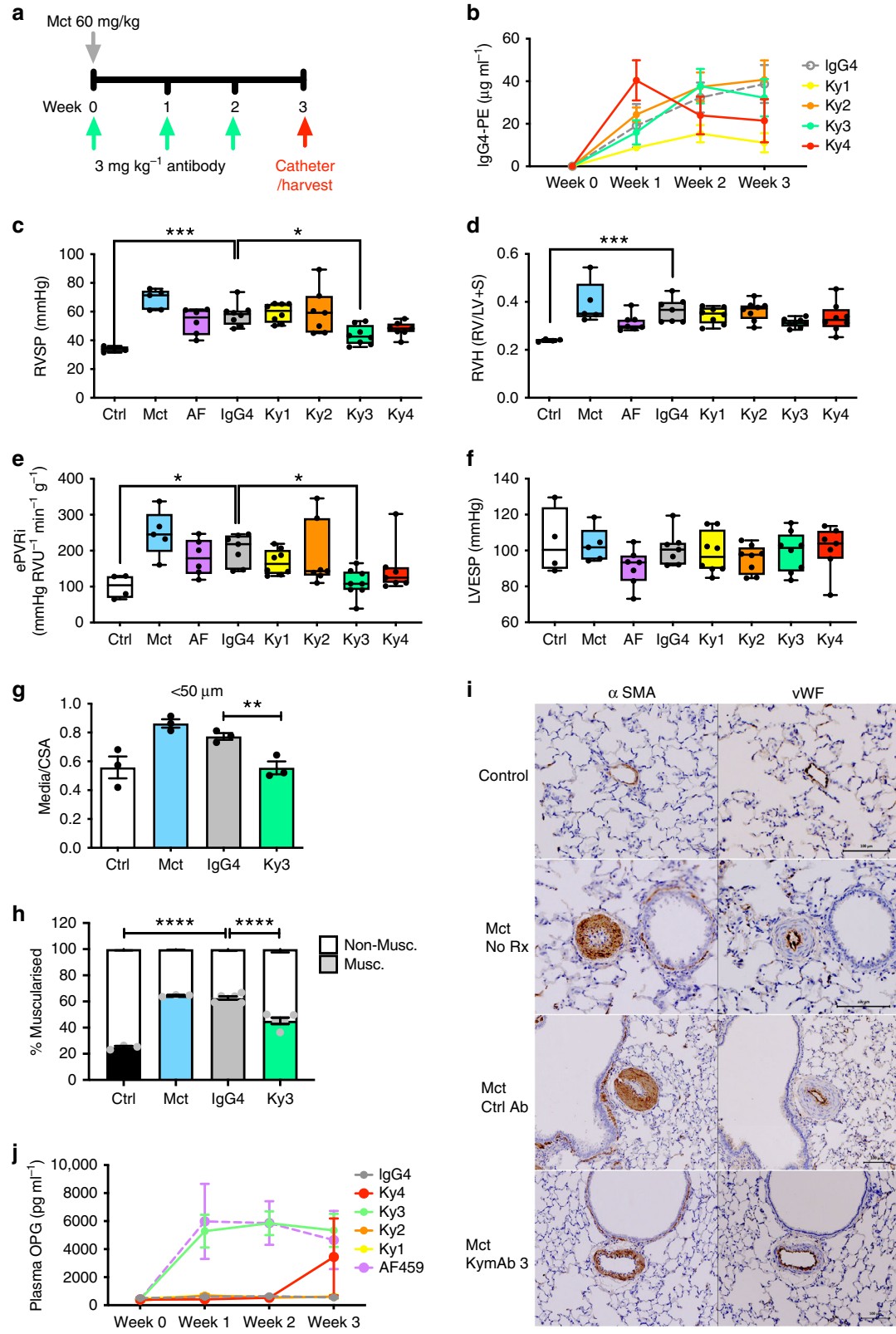

generation of a diverse panel of high affinity anti-human OPG monoclonal antibodies. Various immunisation regimens, including conventional intraperitoneal injections as well as a rapid immunisation at multiple sites (RIMMS) regimes were set up using recombinant human or rat OPG mature peptide sequences fused to human IgG-Fc domains expressed in CHO cells (Supplementary Figure 4). At the end of each regime, secondary lymphoid tissue such as the spleen, and in some cases, the lymph nodes were removed. Tissues were prepared into a single cell suspension and fused with SP2/0 cells by electrofusion to generate stable

hybridoma cell lines. A number of human and mouse OPG cross-reactive antibodies were grouped by their neutralisation profiles and varying ability to block the interaction of OPG with TRAIL and RANKL were identified following the assessment of hybridoma supernatants in a sequential primary and secondary screen cascade using HTRF® (Homogeneous Time-Resolved Fluorescence — see Supplementary Methods) and label-free surface plasmon resonance (SPR). Selected leads were produced in larger quantity in suspension CHO cells and purified as fully human IgG4 PE (human IgG4 Fc region with mutated to amino

**Fig. 6** Human anti-OPG antibody attenuates monocrotaline-induced PAH in rats. Panel (**a**) shows the schema for disease initiation and treatment time course. **b** Plasma concentrations of antibody and IgG. Bar graphs show (**c**) right ventricular systolic pressure (RVSP), (**d**) right ventricular hypertrophy (RVH), (**e**) estimated pulmonary vascular resistance (ePVRi), (**f**) left ventricular end-systolic pressure (LVESP), (**g**) the degree of medial wall thickness as a ratio of total vessel size (Media/CSA), (**h**) relative percentage of muscularised small pulmonary arteries and arterioles in <50 μm vessels. Panel (**i**) shows representative photomicrographs of serial lung sections. Sections were immunostained for α-smooth muscle actin (α-SMA), or von Willebrand factor (vWF). Panel (**j**) shows the circulating plasma levels of OPG. Box and Whisker plots represent the interquartile range (box) with the line representing the median and whisker the full range of the data, each animal is represented by a dot. Ctrl boxes (white, $n = 4$), Mct (blue $n = 5$), AF459 (purple, $n = 6$), IgG (grey, $n = 8$), Ky1 (yellow, $n = 8$), Ky2 (orange, $n = 7$), Ky3 (green, $n = 8$) and Ky4 (red, $n = 7$). * $p < 0.05$, ** $p < 0.01$, *** $p < 0.001$ compared to IgG treated rats following one-way ANOVA followed by Bonferroni's multiple comparisons test. All images are presented at their original magnification ×400, scale bar represents 100 μm

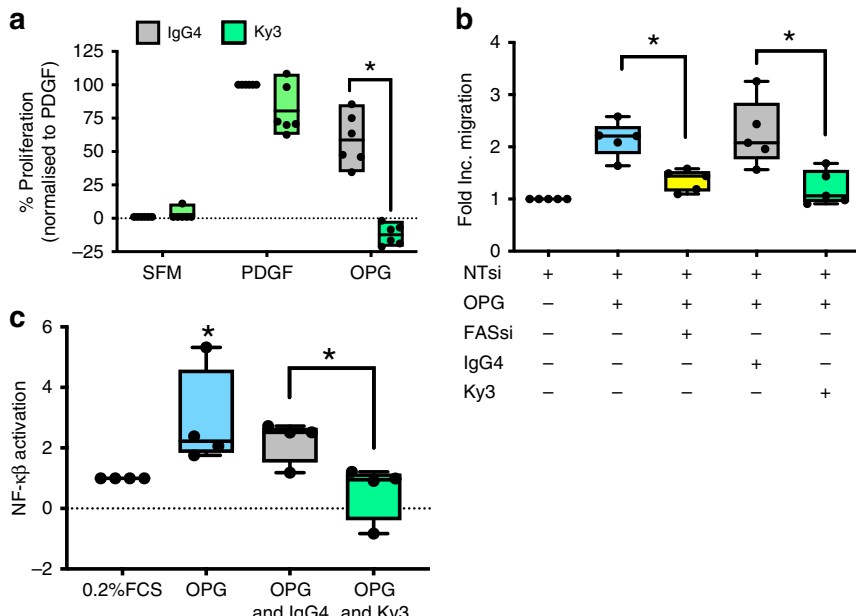

**Fig. 7** Ky3 blocks OPG-induced proliferation, migration and NF-κβ activation. Box and whisker plots shows the inhibition of OPG-induced proliferation (**a**) and migration (**b**) in PASMC stimulated with serum-free media (SFM), PDGF or OPG in the presence of either IgG4 (grey) or Ky3 antibody (green), Fas siRNA (yellow) or non-targeting siRNA (NTsi) (white). Bar graph shows the mean with the error bars showing the standard error o the mean with (**c**) showing the activation of NF-κβ in response to OPG (blue) in the presence of either IgG4 (grey) or Ky3 antibody (green). Box and Whisker plots represent the interquartile range (box) with the line representing the median and whisker the full range of the data, each dot represents an experimental repeat, $n = 6$ (**a**), $n = 5$ (**b**) and $n = 4$ (**c**), * $p < 0.05$ following two-way ANOVA followed by Sidak's multiple comparisons test (**a**), or one-way ANOVA with Bonferroni's multiple comparisons post hoc test (**b&c**)

acids P and E at residues S228 and L235 (EU index) to stabilise the hinge region and remove residual antibody-dependent cell-mediated cytotoxicity) and assessed in in vitro and in vivo studies. The anti-OPG antibodies, KY1–KY4, described in this manuscript are corporate assets, protected by various patents and, as such, are only available through licensing or an MTA, the terms of which will be agreed on a case-by-case basis.

**Pulmonary hypertension phenotyping.** Operators were blinded to treatment groups through the collection and analysis of phenotype data. Echocardiography was performed using the Vevo 770 system (VisualSonics, Toronto, Canada) using either the RMV707B (mice) or RMV710B (rat) scan head. Rectal temperature, heart rate and respiratory rate were recorded continuously throughout the study. Anaesthesia was induced and maintained using isoflurane, sustaining heart rates at 450–500 (mice) and 325–350 (rats) beats per minute (bpm). Rodents were depilated and pre-heated ultrasound gel applied (Aquasonics 100 Gel, Parker Labs Inc., Fairfield, NJ). Right ventricle free wall parameters were collected using M-mode from the right parasternal long axis view. Standard left ventricle parameters were determined using two-dimensional, M-mode and Doppler pulse wave in the short axis view at the level of the papillary muscles. Cardiac output (CO) was derived from flow and annulus diameter at the outflow tract and aortic valve junction, then normalised by body weight. Analysis was performed using Vevo 770 software (v3.0, VisualSonics). All measurements were made during the relevant cardiac cycle phase, avoiding inspiration artefact[12,16].

Following echocardiography and under isoflurane-induced anaesthesia, left and right ventricular catheterisation was performed using a closed chest method via the

right internal carotid artery and right external jugular vein. Pressure volume measurements were collected using the following catheters: PVR-1045 1F (mouse LV), PVR-1030 1F (mouse RV), SPR-838 2F (rat LV) and SPR-847 1.4F (rat RV; Millar Inc.), coupled to a Millar MPVS Ultra and PowerLab 8/30 data acquisition system (AD Instruments Ltd, Oxford, UK). Data were recorded using LabChart v7 software (AD Instruments Ltd) and analysed using PVAN v2.3 (Millar, Houston, TX, USA). Estimated pulmonary vascular resistance (ePVRi) was calculated using the equation (estimated mean pulmonary artery pressure(EmPAP) — left ventricular end — diastolic pressure (LVEDP)/cardiac index)[51]. EmPAP was derived from RVSP, by substituting systolic PAP for RVSP, to give [EmPAP = (0.61 x RVSP) + 2 mmHg][52]. EmPAP was then used in place of mean PAP in the PVRi equation shown above 12. The animals were then humanely killed under anaesthesia and tissues harvested for analysis described below[12,26].

**Right ventricular hypertrophy.** Right ventricular hypertrophy (RVH) was measured by calculating the ratio of the right ventricular free wall weight over left ventricle plus septum weight.

**Immunohistochemistry.** Immediately after harvest, the left lung was perfusion fixed via the trachea with 10% (v/v) formalin buffered saline by inflation to 20 cm of $H_2O$. The lungs were then processed into paraffin blocks for sectioning. Paraffin embedded sections (5 μm) of mouse and rat lung were histologically stained for Alcian Blue Elastic van Gieson (ABEVG) and immunohistochemically stained for α-smooth muscle actin (α-SMA (1:150), M0851, Dako (Agilent), Santa Clara, CA,

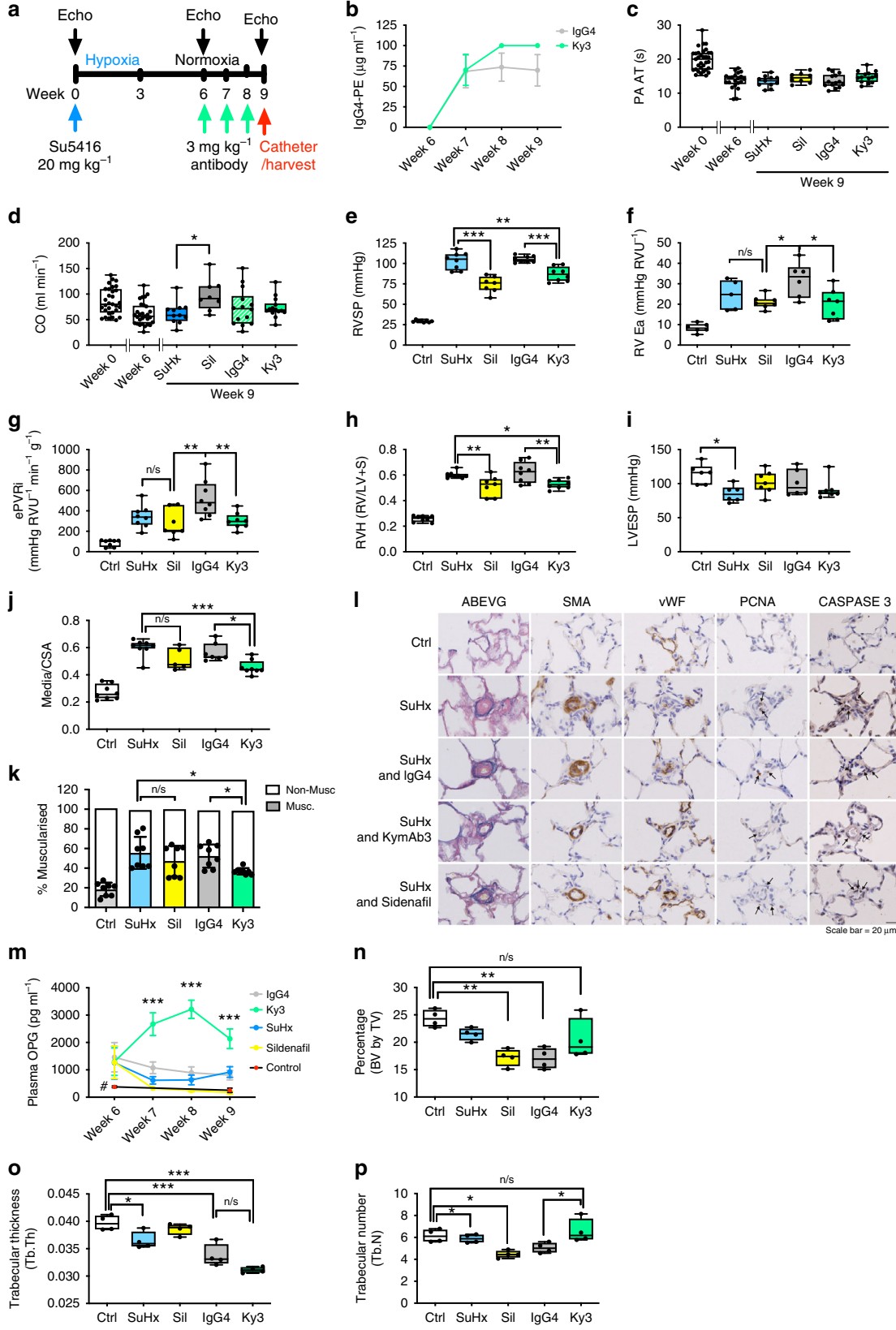

USA); von Willebrand factor (vWF (1:300), A0082, Dako); F4/80 (1:100), ab111101, Abcam, Cambridge, UK); interleukin-6 (IL-6 (1:15), ab6672, Abcam); OPG ((1:50), ab73400, Abcam); TRAIL ((1:100), ab231063, Abcam); Fas ((1:500), ab133619, Abcam) and IκBα ((1:100), ab32518, Abcam). To assess proliferation, slides were stained with a mouse anti-human proliferating cell nuclear antigen antibody (PCNA (1:125), M0879, Dako). In each case a biotinylated secondary

antibody (1:200) was added before an avidin-biotin enzyme complex (Vectastain® Kit, Vector Laboratories, Burlingame, CA, USA) and 3,3′-diaminobenzidine tetrahydrochloride (DAB) substrate. Apoptotic nuclei were detected with a TUNEL assay using a colorimetric DNA fragmentation detection kit (fragEL™, QIA33, Calbiochem®, Merck, Burlington, MA, USA)[12,26], or stained immunohistochemically for cleaved caspase 3 ((1:50), 9661, Cell Signalling Technology, Danvers, MA,

**Fig. 8** Therapeutic delivery of Ky3 attenuates development of established severe SuHx PAH. Panel (**a**) shows the schema for disease initiation and treatment time course. **b** Plasma concentrations of antibody and IgG. Bar graphs show (**c**) Pulmonary Artery Acceleration Time (PA AT), (**d**) cardiac output, (**e**) right ventricular systolic pressure (RVSP), (**f**) right ventricular arterial elastance (RV Ea), (**g**) estimated pulmonary vascular resistance (ePVRi), (**h**) right ventricular hypertrophy (RVH), (**i**) left ventricular end-systolic pressure (LVESP). Bar graphs (**j**) show the degree of medial wall thickness as a ratio of total vessel size (Media/CSA) and (**k**) the relative percentage of muscularised small pulmonary artery and arterioles in < 50 μm vessels. Panel (**l**) shows representative photomicrographs of serial lung sections. Sections were stained for Alcian Blue Elastic van Gieson (ABEVG), immunostained for α-smooth muscle actin (α-SMA), or von Willebrand factor (vWF), proliferating cell nuclear antigen (PCNA) or cleaved Caspase 3. Panel (**m**) shows the circulating level of OPG and quantification of femoral trabecular bone volume (%) (**n**), trabecular thickness (mm) (**o**), trabecular number (mm$^{-1}$) (**p**). Box and Whisker plots represent the interquartile range (box) with the line representing the median and whisker the full range of the data, each animal is represented by a dot, white boxes represent control ($n = 8$), blue (SuHx, $n = 8$), yellow (Sildenafil treated, $n = 7$), grey (IgG4 treated, $n = 8$) and green (Ky3 treated, $n = 8$) rats. * $p < 0.05$, ** $p < 0.01$, *** $p < 0.001$ compared to IgG treated rats following one-way ANOVA with Tukey's multiple comparisons post hoc test. All images are presented at their original magnification ×400, scale bar represents 20 μm

---

USA). Human pulmonary artery and right ventricle histology sections were obtained from patients with IPAH and control lung resection patients from Papworth Hospital (Cambridge, UK) tissue bank and immunohistochemically stained for Fas ((1:100), ADI-AMM-227-E, Enzo Life Sciences, Exeter, UK) and IL-1RAcP ((1:1000), ab8110, Abcam).

**Immunofluorescent staining.** Lung tissue was obtained from chronically hypoxic neonatal calves and normoxic age-matched controls. This neonatal calf model of severe hypoxic pulmonary hypertension has been described previously[53] and includes the development of PA pressure equal to, or exceeding, systemic pressure as well as remarkable PA remodelling with medial and adventitial thickening, resembling that of human neonatal PH. Indirect immunostaining was performed with rabbit polyclonal anti-OPG antibodies ((1:500), Bioss Antibodies, Woburn, MA, USA) followed by biotin-conjugated anti-rabbit secondary antibody ((1:100), Vector Laboratories) and Streptavidin-Alexa-488 ((1:200), Invitrogen, Carlsbad, CA, USA).

**Quantification of pulmonary vascular remodelling.** Images of stained sections were captured using a Zeiss Imager Z2 microscope with an Axiocam 506 colour (brightfield) or MRm (fluorescence) camera with HXP 120 V light source (Carl Zeiss, Oberkochen, Germany). Zen 2 software (Carl Zeiss) was used for image analysis. Pulmonary vascular remodelling was quantified by assessing the degree of muscularisation and the percentage of affected pulmonary arteries and arterioles. For each lung, pulmonary arteries were categorised as either muscularised (i.e. with crescent or complete rings of muscle) or non-muscularised (no apparent muscle) on ABEVG stained sections. Vessels were also divided into sub-groups determined by their external diameter: <50 μm for small arterioles and, additionally where stated, 51–100 and >100 μm for medium arteries. The proportion of muscularised vessels within each sub-group was calculated as a percentage of the total number of vessels. The degree of muscularisation was also determined for each group, and given as the area of positive α-smooth muscle actin staining in the vessel media divided by the total vessel cross-sectional area (media/CSA)[12].

**Quantification of bone structure by microCT.** Femora were scanned on a Skyscan microCT scanner (1172a, Bruker, Belgium) at 50 kV and 200 μA using a 0.5 mm aluminium filter and a detection pixel size of 4.3 μm. Images were captured every 0.7° through 180° rotation and 2x averaging of each bone. Scanned images were reconstructed using Skyscan NRecon software (v. 1.6.8.0) and datasets analysed using Skyscan CT analysis software (v. 1.13.2.1). Trabecular bone was measured over a 1 mm³ volume, 0.2 mm from the growth plate. Trabecular bone volume as a proportion of tissue volume (BV/TV, %), trabecular thickness (Tb. Th, mm), trabecular number (Tb. N, mm$^{-1}$) and trabecular structure model index (SMI) were assessed in this area. Cortical bone was measured over a 1 mm³ volume, 1 mm from the growth plate, and cortical bone volume (C. BV, mm³) assessed in this area.

**Cell culture.** Prior to experimentation, human PASMCs (CC2581; Lonza, Basel, Switzerland) were sub-cultured in SmBM containing SmGM-2 SingleQuot™ Kit supplements and growth factors (Lonza) containing penicillin and streptomycin at 37 °C (5% CO$_2$). Cells were synchronised with growth arrest media (DMEM, 0.2% FBS, penicillin and streptomycin) for 48 h prior to stimulation. All experimentation was conducted at 37 °C with 5% CO$_2$ with cells aged between passage 4–7.

**Proliferation assay.** PASMCs were seeded into 96 well plates ($0.5 \times 10^4$ cells per well) and allowed to adhere for 24 h (37 °C, 5% CO$_2$). Cells were then synchronised with growth arrest media (DMEM, 0.2% FBS, penicillin and streptomycin) for 48 h prior to stimulation. PASMCs were pre-incubated with Fas neutralising antibody (1500 ng ml$^{-1}$, Clone ZB4, Merck) and/or TRAIL neutralising antibody (1500 ng ml$^{-1}$, Clone 75411, R&D Systems), where indicated for 30 min before stimulation with PDGF (20 ng ml$^{-1}$, R&D Systems) or OPG (30 ng ml$^{-1}$, R&D Systems).

Proliferation was assessed after 72 h using the CellTiter-Glo® Luminescent Cell Viability Assay (Promega, Southampton, UK).

**Kinex antibody microarray (KAM).** PASMCs were synchronised with growth arrest media (DMEM, 0.2% FBS, penicillin and streptomycin) for 48 h prior to stimulation. Cells were then stimulated with 0.2% (v/v) FBS (negative), rhOPG (50 ng ml$^{-1}$) and PDGF (20 ng ml$^{-1}$) for 10 and 60 min. Phosphorylation targets were identified from protein lysates by Kinex antibody microarray (Kinexus, Vancouver, Canada). A Z-ratio of ± 1.5 was deemed significant. Uniprot accession codes of proteins were analysed using the Database for Annotation, Visualization and Integrated Discovery (DAVID) functional annotation to generate fold enrichment pathway analysis through the KEGG Pathway Database.

**Western blotting.** PASMCs were stimulated with rhOPG (50 ng ml$^{-1}$) (R&D systems), alongside quiesced cells (negative control) for 10 and 60 min, before lysing. Cell lysates were mixed with sample buffer (Life Technologies, Carlsbad, CA, USA) and sample reducing agent (Life Technologies), denatured by heating and subjected to gel electrophoresis. The membranes were then incubated with primary antibodies against phospho-CDK4, phospho-HSP27, total mTOR, phospho-mTOR (1:500) and GAPDH (1:1000) (Cell Signalling Technology), CDK5 (1:500) (Abcam), or β-actin (1:1000) (Santa Cruz Biotechnology, Heidelberg, Germany). Membranes were then incubated with anti-Rabbit IRDye 800CW and anti-Mouse IRDye 800CW (Li-COR, Lincoln, NE, USA) and signal detection and band density quantification was performed using the LiCOR Odyssey SA system.

**Retrogenix cell microarray.** Identification of OPG human protein binding partners was performed using the Retrogenix Cell Microarray (Sheffield, UK). Optimal binding conditions were first established using syndecan-1 (positive control) and TREM-1 (negative control). HEK293 cells were reverse transfected with expression vectors consisting of one of 2505 human plasma membrane proteins. Cells were treated with 0.5 μg ml$^{-1}$ rhOPG (Peprotech, London, UK), 0.5 μg ml$^{-1}$ anti-OPG (Peprotech) followed by Alexafluor647 anti-goat antibody. Fluorescent images were analysed and quantified using the ImageQuant software (GE) (http://www.retrogenix.com/default.asp).

**Co-immunoprecipitation.** PASMCs were stimulated with rhOPG (500 ng ml$^{-1}$) for 30 min at 37 °C. After stimulation, cells were lysed and the protein lysate concentration determined by a Pierce 660 nm protein assay. Co-immunoprecipitation was then performed using an anti-Fas or Ky3 antibody with human PASMC lysate and recombinant proteins, alongside negative controls, where antibodies were not added. ProteinG sepharose 4 Fast Flow beads (50% slurry) were added to each Co-IP reaction and immune complexes were precipitated. Each Co-IP reaction was then centrifuged and the pellet washed before re-suspending in sample reducing agent (NuPAGE, Life Sciences) with 5% v/v SDS and heating at 95 °C. The supernatant was then analysed by western blotting. Membranes were incubated with goat polyclonal anti-OPG antibody (1:1000) (SC8468, Santa Cruz Biotechnology) or anti-Fas antibody (MA1–7622, Invitrogen) and IRDye 680LT Donkey anti-goat secondary antibody (1:15000) or IRDye 800CW donkey anti-mouse secondary antibody (1:15000) (Li-COR) to detect co-immunoprecipitated OPG. Membranes were scanned using the Li-COR Odyssey Sa system (LiCOR).

**HT1080 apoptosis assay.** HT1080 cells (CCL121; ATCC, USA) were seeded at $5 \times 10^4$ cells per ml in 96 well white walled cell culture plates in EMEM (EBSS) with 2 mM glutamine, 1% non-essential amino acids (NEAA) and 10% foetal bovine serum (FBS) (Life Sciences Ltd, UK). After 24 h, cells were stimulated with OPG 30 ng ml$^{-1}$ alone, or OPG 30 ng ml$^{-1}$ with 1 or 5 ng cross-linked FasL (R&D Systems), 2 nM Fas neutralising Ab (05–338, Merck) or 5 ng ml$^{-1}$ TRAIL (R&D Systems). Apoptosis was measured using a Caspase 3/7 assay (G8091, Promega).

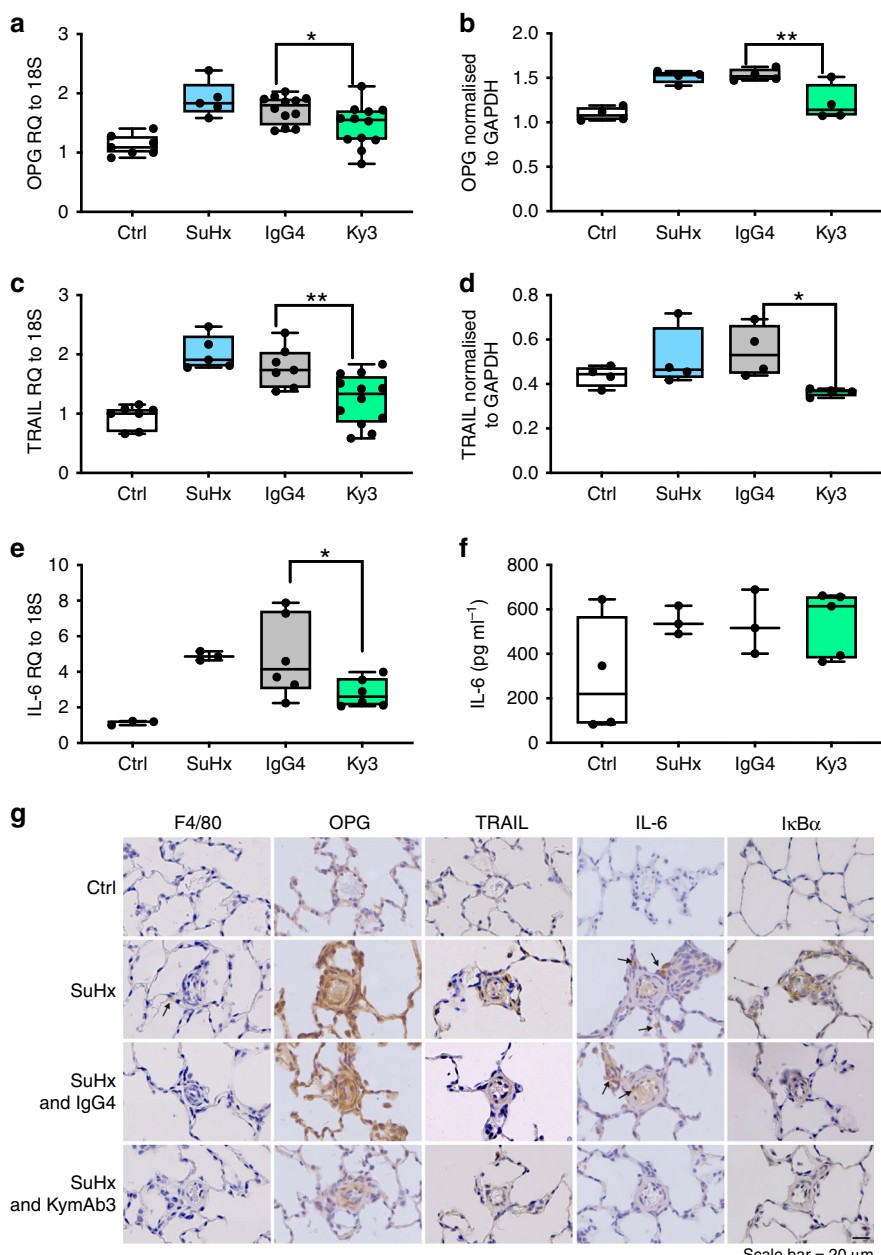

**Fig. 9** *Ky3 reduces tissue expression of IL-6, OPG and TRAIL*. Boxplots demonstrate a significant reduction in whole lung expression of OPG RNA (**a**), OPG Protein (**b**), TRAIL RNA (**c**), TRAIL protein (**d**) and IL-6 RNA (**e**) and plasma protein (**f**). Panel (**g**) shows representative photomicrographs of serial lung sections. Sections were stained for macrophages (F4/80), OPG, TRAIL, IL-6 and IκBα. Box and Whisker plots represent the interquartile range (box) with the line representing the median and whisker the full range of the data, each animal is represented by a dot (Ctrl (white) $n = 7$, SuHx (blue) $n = 5$, IgG (grey) $n = 7$ & Ky3 (green) $n = 12$ animals per group). * $p < 0.05$, ** $p < 0.01$, compared to IgG treated rats using one-way ANOVA followed by Sidak's multiple comparisons test. All images are presented at their original magnification ×400, scale bar represents 20 μm

**Agilent RNA microarray**. mRNA expression profiling was performed using the SurePrint G3 Human Gene Expression 8 × 60 K v2 Microarray according to the manufacturer's instructions (Agilent Technologies, UK). Human PASMCs (Lonza) were stimulated in triplicate with 0.2% FCS (control) or 50 ng ml$^{-1}$ OPG (Peprotech). RNA samples (200 ng) from each condition were labelled and hybridised using standard Agilent protocols. Sample array matrices were scanned on an Agilent Technologies Scanner G2505C using Feature Extraction Software (Agilent Technologies). Loess normalisation and data analysis was performed using the Linear Models for Microarray Data (LIMMA) package[54] in R (http://www.r-project.org/). Data were analysed by two means. (1) A Medline (PubMed) search using term 'pulmonary hypertension' was used to compile a curated list of disease-relevant genes (Supplemental table 5) (39). This list was used to identify PAH related genes differentially regulated in PASMCs between OPG and control samples (BH adjusted $p$-value < 0.05 and log2 FC > 1.2). (2) Signalling Pathway Impact Analysis (SPIA) is an unbiased method that combines over-representation analysis

with a measurement of the perturbation in a pathway to identify signalling networks that are relevant in a given dataset. Full gene expression data (not filtered for PAH relevant genes) were analysed (BH adjusted $p$-value < 0.01) using the SPIA package[25] in R to identify KEGG Pathways[55–57] regulated by OPG.

**Taqman PCR**. PASMCs were stimulated with 0.2% (v/v) FCS (control) or OPG (50 ng ml$^{-1}$) alone or in the presence of Fas antibody (1500 ng ml$^{-1}$) following 30 minute pre-incubation with Fas antibody. After 6 h stimulation, total RNA was extracted using the Direct-zol™ RNA kit (Zymo Research, Irvine, CA, USA). Purified RNA was reverse transcribed with the High Capacity RNA-to-cDNA Kit (Life Technologies). Gene expression was measured by performing TaqMan PCR using Gene Expression MasterMix (Applied Biosystems) for, Cav-1 (Hs00971716_m1), PDGFRa (Hs00998018_m1), TNC (Hs01115665_m1), TRAIL (Hs00921974_m1, Rn0059556_m1, Mn01182929_m1), VEGFA (Hs00900055_m1), VIPR1

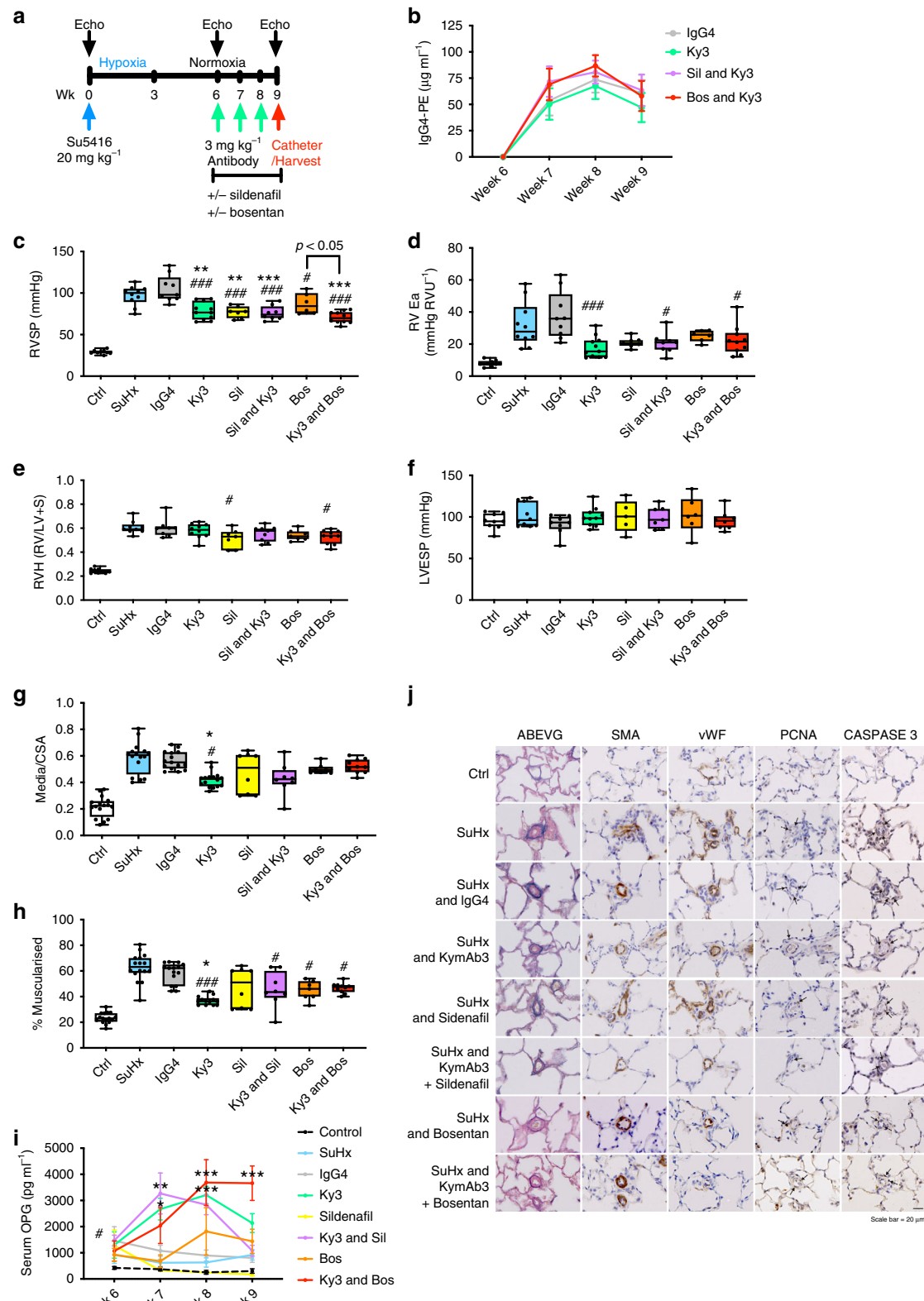

(Hs00270351_m1), Fas (Hs00236330_m1, Rn00685720_m1) and OPG (Mn01205928_m1, Rn00563499_m1) on the 7900HT fast real time PCR system (Applied Biosystems). Gene expression was calculated using the ΔΔ$C_T$ comparative quantification method with 18 S rRNA (Hs03003631_g1) as an endogenous control.

**NF-kB activation assay.** PASMCs were seeded into 96 well plates ($0.5 \times 10^4$ cells per well) and allowed to adhere for 24 h (37 °C, 5% $CO_2$). Cells were then transfected with 100 ng per well inducible NFkB responsive firefly luciferase reporter and constitutively active Renilla construct mixture using the Cignal reporter assay

kit (Qiagen) and Lipofectamine 2000 transfection reagent (Invitrogen) and incubated for 24 h (37 °C, 5% $CO_2$). Media was then renewed in the presence or absence of stimulation with OPG (30 ng ml$^{-1}$, R&D Systems) with or without 1500 ng ml$^{-1}$ of Ky3 or control IgG4 antibodies. Luciferase activity was detected following 48 h stimulation using Dual-Glo luciferase assay system (Promega).

**Human tissue.** Experimental procedures using human tissues or cells conformed to the principles outlined in the Declaration of Helsinki. Papworth Hospital ethical review committee approved the use of the human tissues (Ethics Ref 08 -H0304–56

**Fig. 10** Ky3 and standard of care vasodilator therapy combination attenuates severe PAH. Panel (**a**) shows the schema for disease initiation and treatment time course. **b** Plasma concentrations of antibody and IgG. Boxplots show (**c**) right ventricular systolic pressure (RVSP), (**d**) right ventricular arterial elastance (RV Ea), (**e**) right ventricular hypertrophy (RVH), (**f**) left ventricular end-systolic pressure (LVESP), (**g**) degree of medial wall thickness as a ratio of total vessel size (Media/CSA) and (**h**) the relative percentage of muscularised small pulmonary arteries and arterioles in < 50 μm vessels. Graph (**i**) shows the circulating level of OPG, and panel (**j**) shows representative photomicrographs of serial lung sections. Sections were stained for Alcian Blue Elastic van Gieson (ABEVG), immunostained for α-smooth muscle actin (α-SMA), or von Willebrand factor (vWF), proliferating cell nuclear antigen (PCNA) or cleaved Caspase 3. Box and Whisker plots represent the interquartile range (box) with the line representing the median and whisker the full range of the data, each animal is represented by a dot, white boxes represent control ($n = 9$), blue (SuHx, $n = 10$), grey (IgG4 treated, $n = 9$) and green (Ky3 treated, $n = 11$), yellow (sildenafil treated, $n = 7$), purple (sildenafil & Ky3 treated, $n = 8$), orange (bosentan treated, $n = 6$) and red (bosentan & Ky3 treated, $n = 10$) rats. # $p < 0.05$, ## $p < 0.01$, ### $p < 0.001$ compared to IgG, *$p < 0.05$, ** $p < 0.01$, *** $p < 0.001$ compared to SuHx treated rats using one-way ANOVA followed by Sidak's multiple comparisons test. All images are presented at their original magnification ×400, scale bar represents 20 μm

þ 5) and informed consent was obtained from all subjects Sections of formalin-fixed lung and right ventricle from patients with IPAH or unused donors were stained for Fas ((1:500), ab133619, Abcam) and OPG ((1:50), ab73400. In each case a biotinylated secondary antibody (1:200) was added before an avidin-biotin enzyme complex (Vectastain® Kit, Vector Laboratories, Burlingame, CA, USA) and 3,3′-diaminobenzidine tetrahydrochloride (DAB) substrate.

**Statistics.** Statistical analysis was performed using either a one-way ANOVA or two-way ANOVA followed by Sidak's multiple comparisons test or Bonferroni's multiple comparisons test. When there were only two groups, unpaired t-tests were used. $P < 0.05$ was deemed statistically significant (Prism 8.0.2 for Macintosh, Graphpad Software).

**Study approval.** All animal experiments were approved by the University of Sheffield Project Review Committee and conformed to the UK Home Office ethical guidelines.

**Reporting summary.** Further information on research design is available in the Nature Research Reporting Summary linked to this article.

## Data availability

The data that support the findings of this study are available from the corresponding author upon reasonable request. The source data underlying Fig. 3a is available from the Gene Expression Omnibus (GEO), GSE137886. The data for all other figures are provided as a Source Data file.

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

## Acknowledgements

Funding for this study was provided by a British Heart Foundation Senior Basic Science Research Fellow (FS/13/48/30453 and FS/18/52/33808, AL), Medical Research Council Career Development Award (G0800318, AL); Medical Research Council Confidence In Concepts (MC/PC12022), Medical Research Council Developmental Pathway Funding Scheme (MR/L023040/1), British Heart Foundation Clinical Research Training Fellowship (FS/08/061/25740, AGH); Medical Research Council Clinical Research Training Fellowship (MR/K002406/1) and Wellcome Trust Clinical Research Career Development Fellowship (206632/Z/17/Z), AMKR, British Heart Foundation Intermediate Clinical Fellowship (FS/18/13/33281, AART), National Institute for Health Research Sheffield Cardiovascular Biomedical Research Unit (NA/JP/DC); Cambridge National Institute of Health Research Biomedical Research Centre.

## Author contributions

N.D.A. helped design and performed experiments, and helped write the manuscript. S.D. helped design and performed experiments, and helped write the manuscript; J.A.P. helped design and performed experiments, and helped write the manuscript; J.C. helped design and performed experiments, and helped write the manuscript; A.T.B. helped design and performed experiments, and help prepare the manuscript; L.E.W. helped design and performed experiments, analysed data and helped write the manuscript; J.I. helped design and performed experiments, performed statistical analysis of microarray data, and helped write the manuscript; H.C. helped performed experiments; L.R. performed experiments; V.G. helped design and performed experiments; M.M. helped design and performed experiments; A.M.K.R. helped design and performed experiments, performed statistical analysis of microarray data, and helped write the manuscript. H.K. Help perform experiments; A.G.H. helped performed experiments, and helped write the manuscript. M.G.F. helped design and performed experiments, and helped write the manuscript. A.A.R.T. helped design and performed experiments, and help write the manuscript; H.R.E. helped design and performed experiments, and helped write the manuscript; M.S. proved human sections and performed experiments; N.W.M. supplied human sections and RNA for analysis; C.M.N. helped write the manuscript; D.C.C. helped write the manuscript; M.K.B.W. helped write the manuscript; K.R.S. supplied supportive data, access to samples for analysis and helped write the manuscript; P.B-W. helped design and performed experiments, and helped write the manuscript; D.G.K. helped write the manuscript; S.E.F. helped design experiments, and helped write the manuscript; A.L. had the original idea, helped design and performed experiments, and wrote the manuscript.

## Competing interests

A.L./The University of Sheffield has been granted intellectual property around the area of targeting OPG for the treatment of PAH (GB2510524 / US9334327 / JP2014532637) and is a founding Director of PH Therapeutics Ltd, a University of Sheffield Spin-out company. J.C., V.G., M.M. and P.B-W. are employees of Kymab Ltd and hold share options in the company. Kymab has filed intellectual property (GB1701416.8) around the characteristics of a therapeutic human anti-osteoprotegerin antibody with A.L. named as an inventor. The remaining authors declare no competing interests.
