## [Peer Review File · Nature Communications]

Reviewers' comments:

Reviewer #1 - expert in PAH (Remarks to the Author):

The authors report that bone marrow-derived osteoprotegerin contributes to pulmonary vascular remodeling associated with pulmonary hypertension (PH). The authors show that mice lacking osteoprotegerin in their bone marrow display significantly reduced serum levels of osteoprotegerin and are protected from developing PH when compared to wild type or mice lacking osteoprotegerin in tissues. They also show that treatment with a novel human antibody targeting osteoprotegerin prevents or reverses the development of experimental PH in rodents. They have used the Paigen high fat, high cholate containing diet (HFD) fed ApoE ^{-/-} mouse model as well as the Sugene+hypoxia and monocrotaline rat models. Finally, they report that osteoprotegerin promotes cell survival, pro-migratory and pro-proliferative signalling in pulmonary arterial smooth muscle cells (PASMCs) in vitro through binding to Fas receptor.

Some findings are interesting but several are insufficient to draw firm conclusions. Moreover, this study remains incremental when compared with the authors previously published data.

MAJOR CONCERNS:

1. No justification is given for why these particular models are used and their limits are not discussed. In particular, the animal model used in Figure 1 is not a well-recognized animal model to study PH. Due to the important variation in RVSP values between the 4 mice (figure1A), I would recommend to not overinterpreting the experimental results.
2. The beneficial effect of treatment with the identified antibody targeting osteoprotegerin Ky3 are relatively weak as compared to the low dose of sildenafil used in the Sugene+hypoxia rat model (Figure 7). Since the authors found that sildenafil has no significant effect on pulmonary vascular remodeling, co-treatment of rats with sildenafil and Ky3 are requested.
3. What is antigenic determinant of Ky3 and why the other antibodies tested are inefficient? The ability of Ky3 to block the binding of osteoprotegerin to Fas receptor is requested.

4. Because of the small number of samples (ranging from n=2 (Fig2) to n=8), the data in all graphs have to be presented in a dot plot graph with medians (with range or interquartile range) rather than means. Furthermore, there are several inconsistencies between the number of animal used and the number of values indicated in the different panels (for example in Figure 1). Please indicate the statistical test used to show significance in all panels.

5. No significant treatment effect was noted in the monocrotaline rat model with Ky3, but the reasons for these negative results were not explored. Levels of osteoprotegerin, and phosphorylation states of ERK ½ and CDK4/5 could be determined in these monocrotaline-injected rat treated or not with Ky3 and compared to the Sugeng+hypoxia rat model?

6. The authors state that these negative results obtained in the monocrotaline model could be due to the particularly advanced and severe phenotype at initiation of treatment, however patients are often diagnosed at late stages in PH. This notion has to be validated perhaps by lowering the dose of monocrotaline administered in these animals.

7. The in vitro studies are weak. Are these effects dose-dependent? Experiments showing that osteoprotegerin can attenuate the PDGF-induced migration and proliferation are requested. The use of siRNA or antagonist of Fas receptor and or TRAIL is needed.

8. I would recommend consolidating the mRNA expression profiling using more samples or different doses of osteoprotegerin. It is indicated in the legend 3A: "n=3 for pooled triplicate samples". In addition, these identified target genes and signaling pathway could be validated in vivo using the rodent lung tissues in the different animal models using co-immunostaining.

9. Elevated osteoprotegerin levels have been found to increase apoptosis of endothelial progenitor cells (EPCs) by induction of oxidative stress (Kim JY et al., Arthritis Rheum. 2013). As indicated by the authors (page 5), however, previous studies have implicated circulating mesenchymal progenitor cells such as EPCs in the pathogenesis of PH. Why this paradox? Analysis of the EPC population in these rodents is required to clarify this point.

10. Tnfrsf11B, the gene encoding osteoprotegerin, is an established HIF target gene (Wu et al., Genes & Dev. 2015). This point has to be taken into account when analyzing the data regarding its upregulation in the sugeng+hypoxia and the hypoxic neonatal calf models of PH. In addition, how do the authors explain the fact that Sugeng alone increases the level of osteoprotegerin (figure S1)?

11. Experiments with Wild-type (wt) littermate mice are missing.

MINOR CONCERNS:

1. Statistical signs are missing in Figure 3E.

Reviewer #2 - expert in therapeutic antibodies (Remarks to the Author):

The authors describe a novel therapeutic antibody strategy to neutralize OPG for pulmonary hypertension. The manuscript is technically sound, but requires a few additional details and changes prior to publication.

1. The authors state their antibodies are high affinity. Please provide SPR or other data to support this.
2. Also, what is the affinity for rat OPG? The authors suggest their in vivo studies might not reflect the full potential in humans due to incomplete homology. Is the mAb more potent against human OPG?
3. How were the four mAbs chosen? Do they block OPG binding to TRAIL or RANKL equally well? Please provide additional in vitro characterization. Do they block the interaction with Fas?
4. Related to #1-3, what antibody property best correlates with in vivo efficacy?
5. Please define what IgG4 PE means and why IgG4 was chosen.

Reviewer #3 - expert in osteoprotegerin (Remarks to the Author):

The manuscript entitled "A novel therapeutic antibody targeting osteoprotegerin attenuates severe experimental pulmonary hypertension" addresses OPG as therapeutic target in PAH. The authors report a large amount of data obtained in vitro and in vivo in different rodent models of PAH, overall supporting the potential of a therapeutic strategy based on an anti-OPG antibody.

The results are of great interest since innovative strategies for PAH are needed. Moreover, interesting mechanistic insights have been included such as the role of Fas receptor in mediating the effects of OPG in PAH.

I have the following comments:

- since OPG has a pro-inflammatory activity on the vasculature, I was wondering if the authors have assessed the effects of Ky3-treatment on the pro-inflammatory status related to PAH in Ky3-treated animals versus controls (e.g. tissue infiltration of inflammatory cells around the plexiform lesions or expression of inflammatory mediators such as IL-6 or modulation of circulating serum pro-inflammatory chemokines/cytokines)?

- in the in vivo experiments Ky3-treated animals show increased circulating OPG levels compared to controls. The authors hypothesize that it might be due to retention of Ky3-bound-OPG in the circulating system rather than allowing access to vessel wall. Are there evidences of a reduced OPG staining in the tissues of Ky3-treated animals compared to controls in the different PAH models?

How were the circulating levels of OPG in the experiment showed in Figure 1 where a polyclonal anti-OPG antibody was used? The complete kinetics of the circulating levels of OPG should be shown in all in vivo experiments for both controls and treated animals.

I was wondering if the authors have excluded possible cross-reactivity in the assay used to quantify circulating OPG.

- since OPG is the decoy receptor for TRAIL and that TRAIL is involved as well in vascular physiopathology and in endothelial cell biology with conflicting results and implications, do the authors have any information on the modulation of TRAIL due to Ky3 treatment (in terms of soluble TRAIL and/or transmembrane isoform)?

As minor points, editing issues should be addressed such as:

- indication of the statistical significance in figures and legends
- the generation of chimeric mice is missing in the material and methods section
- Figure S3_panel B: there is only one line-graph
- the sentence describing figure 1D is not clear (see results, page 2, second paragraph)
- genes nomenclature
- the PAAT acronym is not mentioned in full in the manuscript

- in Figure 1F the indications of ABEGV and a-SMA are missing

Thank you for reviewing the above manuscript. Below is a point-by-point response to the comments/queries raised with the original reviewers' comments in *italics*, our response in plain text and highlighted changes in the revised manuscript text in **red** below, and in the revised manuscript.

Reviewers' comments:

Reviewer #1 - expert in PAH (Remarks to the Author):

The authors report that bone marrow-derived osteoprotegerin contributes to pulmonary vascular remodeling associated with pulmonary hypertension (PH). The authors show that mice lacking osteoprotegerin in their bone marrow display significantly reduced serum levels of osteoprotegerin and are protected from developing PH when compared to wild type or mice lacking osteoprotegerin in tissues. They also show that treatment with a novel human antibody targeting osteoprotegerin prevents or reverses the development of experimental PH in rodents. They have used the Paigen high fat, high cholate containing diet (HFD) fed ApoE -/- mouse model as well as the Sugem+hypoxia and monocrotaline rat models. Finally, they report that osteoprotegerin promotes cell survival, pro-migratory and pro-proliferative signalling in pulmonary arterial smooth muscle cells (PASMCs) in vitro through binding to Fas receptor.

Some findings are interesting but several are insufficient to draw firm conclusions. Moreover, this study remains incremental when compared with the authors previously published data.

We strongly argue against the 'incremental' comment made by the reviewer. This manuscript builds from an observation reported in 2008 to provide new and novel mechanistic data and a new drug! There are no published data on therapeutically targeting OPG, this is the first description of anti-OPG antibody therapeutic approach in any disease. Our published work on OPG reported heightened expression in tissue and serum but together these data are anecdotal and give no evidence that therapeutically targeting OPG is a viable option for the treatment of PAH. We present here the first evidence that OPG can be targeted by a novel human therapeutic antibody and demonstrate efficacy in preclinical models. This is similar to the reporting of other novel therapeutic targets from translational pre-clinical studies. For example, there was over 12 years between the discovery of the importance of BMPR2 heterozygous mutations in PAH before this was therapeutically targeted by either FK506 (Spiekerkoetter et al JCI 2013) or BMP9 (Long et al Nature Medicine 2015). In addition to validating the OPG as a novel drug target in this revised manuscript, which we now also demonstrate this on top of 'standard of care' vasodilator treatment as requested. This manuscript also identifies a new cell surface receptor for OPG (Fas) and provides evidence that in addition to the previously reported OPG-TRAIL protein-protein interaction, OPG induces TRAIL expression in PASMCs. This further supports our previous work (Hameed et al J Exp Med 2012) demonstrating that tissue expressed TRAIL is a key driver of pulmonary vascular remodelling, and now identifies OPG as a key driver of this pathway. We demonstrate that sequestration of OPG, even in established disease with an antibody is a

viable therapeutic option, that is distinct of current standard of care treatments, all of which are novel findings.

MAJOR CONCERNS:

1. No justification is given for why these particular models are used and their limits are not discussed. In particular, the animal model used in Figure 1 is not a well-recognized animal model to study PH. Due to the important variation in RVSP values between the 4 mice (figure 1A), I would recommend to not overinterpreting the experimental results.

We agree entirely that with each individual model there are limitations that can lead to overinterpretation. This is precisely why we have taken the approach of using multiple models in previous studies (Hameed et al, J Exp Med 2012 & Rothman et al JCI 2016) and this study. We utilise several animal models and at least two species. We present data incorporating knock-out mice in 2 different models, the fat fed mice in Fig 1 and SuHx in Fig S1, as well as monocrotaline (Fig S4, S5 and Fig 6) and SuHx (Fig 8-10) rats. Throughout all of these studies, the data consistently reaffirm the phenotype and conclusion that loss of, or blockade of, OPG attenuates PAH pathogenesis. We agree that there is a spread in RVSP from the 4 individual ApoE^{+/-} mice on high fat data, we show the data point for every mouse/rat used in all studies. Concerns about variability and should be countered with the consistency of message through out all the models. Furthermore, we have repeated many of the models in more than one instance, e.g. for the particular example of Fig1A highlighted by the reviewer, the data in Fig1I (the yellow bar) is distinct from data from an additional 6 mice run at a different time but utilising the same model. These data recapitulate the phenotype from Fig 1A.

We also point out that we included a section on limitations of the models in the final paragraph of the manuscript **“Our study was initially limited by the lack of availability of monoclonal anti-human OPG antibodies with cross-reactivity to rat but we overcame this by generating a suite of human monoclonal antibodies that included Ky3. Although there are limitations of each rodent model of PH used in this study, the utilisation of multiple models, each with different characteristics, combined with human data circumvent these concerns. Furthermore, the efficacy demonstrated here may not reflect the full potential effect in humans due to incomplete homology between human and rat proteins.”**

2. The beneficial effect of treatment with the identified antibody targeting osteoprotegerin Ky3 are relatively weak as compared to the low dose of sildenafil used in the Sugen+hypoxia rat model (Figure 7). Since the authors found that sildenafil has no significant effect on pulmonary vascular remodeling, co-treatment of rats with sildenafil and Ky3 are requested.

We regret that the reviewer felt the therapeutic effect of our antibody was relatively weak. In the context of the severity of disease that we present (RVSP >100 mmHg), these therapeutic

effects (reduce of ~25 mmHg) are at least equivalent to those observed in other studies where where the haemodynamics of the untreated rats are much lower e.g. RVSP ~70 mmHg in Spiekerkoetter et al JCI 2013 and RVSP ~60 mmHg in Long et al Nature Medicine 2015. The dose of Sildenafil (50 mg/kg/day in chow) is in the range of that previously used in the literature (Schermuly RT et al. Chronic sildenafil treatment inhibits monocrotaline-induced pulmonary hypertension in rats. *Am J Respir Crit Care Med.* 2004;169:39–45.) and as highlighted above is in the region of 50-fold higher than the dose given to patients with PAH (in order to show efficacy). Extrapolating dose and effect from rodent to man is notoriously difficult and there is no evidence that when given to rats at a human dose equivalent there are any therapeutic effects. In our opinion demonstrating that there is a differentiating mechanism of action i.e. anti-proliferative vs vasodilation is the most important output from a study, however, we concede that there is a growing voice within sectors of the community to perform the combination experiments suggested by the reviewer.

We now report that having conducted a large study with an additional 100 rats, and a separate batch of Ky3 antibody in combination with either sildenafil (PDE5i) as requested, or bosentan (ERA), that the mechanism of action is preserved, the efficacy is maintained in the presence of standard of care drug, and that there is no detrimental effect on existing therapy i.e. there is no interference. Furthermore, we report further reduction in RVSP with the combination of the Ky3 and bosentan compared to either treatment alone. We have therefore included these new data as Figure 10 in the revised manuscript with the new text below included.

Finally, Ky3 antibody (3 mg/kg/week) was then tested in comparison, and combination with, sildenafil (50 mg/kg/day) or bosentan (60 mg/kg/day) treated rats exposed to SuHx, with IgG4 treatment as a control (Fig 10A). There was no effect of either sildenafil or bosentan on the levels of circulating Ky3 as measured by IgG4 luminex assay (Fig 10B). Treatment of SuHx rats with sildenafil, bosentan or Ky3 resulted in a comparable reduction PAH phenotype (Fig 10C-H). Ky3 in combination with bosentan resulted in a significant further reduced RVSP compared to bosentan alone (Fig 10C). Sildenafil treated rats only demonstrated a reduction in pulmonary vascular remodelling when also receiving Ky3 (Figure 10H). As previously demonstrated rats treated with Ky3 had increased circulating levels of OPG (Fig 10I). Caspase 3 and PCNA staining identified an increase in apoptosis and decrease in proliferation within the small remodelled pulmonary arterioles in the lungs of rats treated with Ky3 when compared to either sildenafil or bosentan alone (Fig 10J).

3. What is antigenic determinant of Ky3 and why the other antibodies tested are inefficient? The ability of Ky3 to block the binding of osteoprotegerin to Fas receptor is requested.

We apologise for not providing more information within the methods section of the manuscript. We have subsequently added text to the methods to clarify the antigenic generation and epitope mapping of the generated antibodies. Changes are highlighted in red below and in the revised manuscript.

“KyMouse™ system of genetically engineered mice containing a large number of human immunoglobulin genes 25 was used for the generation of a diverse panel of high affinity anti-human OPG monoclonal antibodies. Various immunisation regimens, including conventional intraperitoneal injections as well as a rapid immunisation at multiple sites (RIMMS) regimes were set up using recombinant human or rat OPG mature peptide sequences fused to human IgG-Fc domains expressed in CHO cells (Fig S3). At the end of each regime, secondary lymphoid tissue such as the spleen, and in some cases, the lymph nodes were removed. Tissues were prepared into a single cell suspension and fused with SP2/0 cells by electrofusion to generate stable hybridoma cell lines. A number of human, mouse and rat cross-reactive OPG antibodies were grouped by their neutralisation profile ability to block the interaction of OPG with TRAIL and RANKL were identified following the assessment of hybridoma supernatants in a sequential primary and secondary screen cascade using HTRF® (Homogeneous Time-Resolved Fluorescence) and label-free surface plasmon resonance (SPR). Selected leads were produced in larger quantity in suspension CHO cells and purified as fully human IgG4 PE (human IgG4 Fc region with mutated to amino acids P and E at residues S228 and L235 (EU index) to stabilise the hinge region and remove residual antibody-dependent cell-mediated cytotoxicity) and assessed in *in vitro* and *in vivo* studies (Fig S3).”

Both Ky3 and Ky4 both showed some therapeutic effects *in vivo* and also have the highest apparent KD for rat OPG. We believe that this may explain their better efficiency over other antibodies in the initial *in vivo* model.

With regard to the demonstration Ky3 blocking the binding of osteoprotegerin to Fas receptor we have included new functional data demonstrating that Ky3 inhibits the OPG-FAS induced changes in NF- κ B activation, migration and proliferation (Fig 7). Notably Fas siRNA phenocopies the inhibitory effect of Ky3 on OPG-induced migration. The following text and figure have been added to the revised manuscript.

Ky3 inhibits OPG-induced proliferation, migration and NF- κ B activation.

Once Ky3 was identified as the lead candidate antibody for further development we confirmed that Ky3 inhibited OPG-induced proliferation (Fig 7A) and migration (Fig 7B) in human PSMC *in vitro*. Having previously identified an NF- κ B response to OPG (Fig 3E) we investigated, and show that Ky3 inhibits this activation (Fig 7C).

Fig. 7

*Fig. 7 – Ky3 blocks OPG-induced proliferation, migration and NF- κ B activation. Box and whisker plots shows the inhibition of OPG-induced proliferation (A) and migration (B) in PASCs stimulated with serum-free media (SFM), PDGF or OPG in the presence of either IgG4 or Ky3 antibody, Fas siRNA or non-targeting siRNA (NTsi). Bar graph (C) shows the activation of NF- κ B in response to OPG in the presence of either IgG4 or Ky3 antibody. Box and Whisker plots represent the interquartile range with the line representing the median, n=6 (A), n=5 (B) and n=4 (C), * = p<0.05 following two-way ANOVA followed by Sidak's multiple comparisons test (A), or one-way ANOVA with Bonferroni's multiple comparisons post-hoc test (B&C).*

4. Because of the small number of samples (ranging from n=2 (Fig2) to n=8), the data in all graphs have to be presented in a dot plot graph with medians (with range or interquartile range) rather than means. Furthermore, there are several inconsistencies between the

number of animal used and the number of values indicated in the different panels (for example in Figure 1). Please indicate the statistical test used to show significance in all panels.

We apologise for the oversight in not including data on a few parameters from all the animals studied. This has now been corrected in the new revised version of Fig 2. We has also converted all the *in vivo* data to be presented as Box and Whisker graphs with all data points included, median and interquartile interquartile range. We apologise for the omission of the statistical analysis used in all the figure legends this has also been corrected in this revised version of the manuscript.

5. No significant treatment effect was noted in the monocrotaline rat model with Ky3, but the reasons for these negative results were not explored. Levels of osteoprotegerin, and phosphorylation states of ERK1/2 and CDK4/5 could be determined in these monocrotaline-injected rat treated or not with Ky3 and compared to the Sugen+hypoxia rat model?

The reviewer is correct that we did not see a significant effect on haemodynamics but we did see a significant effect on pulmonary vascular remodelling. We do not view these as negative results although appreciate that the original manuscript may have had that ‘tone’. We rationalise the difference in effect to differences in treatment duration (2 weeks in Mct vs 3 weeks in SuHx) and apparent severity of disease in the particular instance of the Mct model. To try to emphasise this we have added the following text to the manuscript.

In the Mct model we also observed a significant reduction in pulmonary vascular remodelling (Fig S5 J&K) with only 2 weeks of Ky3 treatment but this did not alter the haemodynamic profile (Fig S5). We proposed that this was due to the shorter treatment duration and, particularly advanced/severe phenotype in this instance of the model.

It is important to recognise that both the drivers, and time course of PAH development and right heart failure in the monocrotaline versus Sugen5416 plus hypoxia are different, and are performed in different strains of rat. While we try our best to mitigate these limitations (by performing multiple models) they do not fully recapitulate human disease pathogenesis. The time course of expression of proteins, particularly phosphorylated proteins e.g. ERK1/2 and CDKs that turn over rapidly are difficult to extrapolate from *in vitro* to *in vivo*. We tried but could not reliably pick up a consistent signal in these rapidly changing proteins. We have however added a new data from the Ky3 SuHx study demonstrating that Ky3 reduces tissue expression key PAH and OPG regulated genes and protein. The follow text and figure has been added to the revised manuscript, and levels of circulating OPG are now included in all animal studies.

Treatment of SuHx rats with Ky3 reduces tissue expression of IL-6, OPG and TRAIL.

To demonstrate that the therapeutic effects of Ky3 treatment in the SuHx rat model were associated with reduced OPG signalling, we examined the expression of OPG and identified downstream mediators in the lung tissue. Despite the increase in circulating levels of OPG (Fig 8M), Ky3 treatment resulted in a significant reduction in OPG RNA (Fig 9A) and protein within whole lung lysates (Fig 9B), and within remodelled pulmonary arterioles by IHC (Fig 9G). Similarly, levels of TRAIL were also decreased at RNA (Fig 9C) and protein level (Fig 9D&G). Treatment with Ky3 was also associated with a reduction in inflammation within the lung as shown by IL-6 RNA expression (Fig 9E) and $I\kappa\beta\alpha$ protein (Fig 9G), although there was no effect on total circulating levels of IL-6 (Fig 9F).

Fig. 9

Fig. 9 – Ky3 treatments reduces vascular expression of OPG, TRAIL, and inflammatory mediators in SuHx rats. Boxplots demonstrate a significant reduction in whole lung expression of OPG RNA (A), OPG Protein (B), TRAIL RNA (C), TRAIL protein (D) and IL-6 RNA (E) and plasma protein (F). Panel (G) shows representative photomicrographs of serial lung sections. Sections were stained for macrophages (F4/80), OPG, TRAIL, IL-6 and $\kappa\beta\alpha$. Box and Whisker plots represent the interquartile range with the line representing the median, each animal is represented by a dot (Ctrl n=7, SuHx n=5, IgG n=7 & Ky3 n=12 animals per group). *= $p < 0.05$, **= $p < 0.01$, compared to IgG treated rats using one-way ANOVA followed by Sidak's multiple comparisons test. All images are presented at their original magnification x400, scale bar represents 20 μm .

6. *The authors state that these negative results obtained in the monocrotaline model could be due to the particularly advanced and severe phenotype at initiation of treatment, however patients are often diagnosed at late stages in PH. This notion has to be validated perhaps by lowering the dose of monocrotaline administered in these animals.*

As with our response to point 5 above, we do not view the results from the monocrotaline model as negative as we observed a significant reduction in pulmonary vascular remodelling. We agree that patients currently present at an advanced point in disease but as highlighted above difference in the response between the two models may reflect more on the shorter duration treatment rather than just severity. Given the request that this reviewer also made repeat the SuHx in combination with standard of care PAH therapies we prioritised that repeated the model with a lower dose of monocrotaline.

7. *The in vitro studies are weak. Are these effects dose-dependent? Experiments showing that osteoprotegerin can attenuate the PDGF-induced migration and proliferation are requested. The use of siRNA or antagonist of Fas receptor and or TRAIL is needed.*

We apologise for not including more in vitro data in the first submission. We have included additional *in vitro* data (Fig 7) demonstrating that Ky3 blocks OPG induced changes in NF- κB activation, migration and proliferation. Notably as requested Fas siRNA phenocopies the inhibitory effect of Ky3 on OPG-induced migration. The following text and figure have been added to the revised manuscript. - See response to point 3 above.

We are slightly confused by the request for “*Experiments showing that osteoprotegerin can attenuate the PDGF-induced migration and proliferation*”. Both OPG and PDGF induce proliferation and migration and we would not expect OPG to attenuate PDGF-induced proliferation or migration. Indeed it is widely reported that PDGF can stimulate OPG (e.g. Zhang J, Fu M, Myles D, Zhu X, Du J, Cao X, Chen YE and PDGF induces osteoprotegerin expression in vascular smooth muscle cells by multiple signal pathways. FEBS Lett. 2002;521:180–184). We believe that OPG is downstream of PDGF but can feedback through PDGF receptor regulation.

8. *I would recommend consolidating the mRNA expression profiling using more samples or different doses of osteoprotegerin. It is indicated in the legend 3A: “n=3 for pooled triplicate samples”. In addition, these identified target genes and signaling pathway could be validated in vivo using the rodent lung tissues in the different animal models using co-immunostaining.*

The dose of OPG is based on our original description of OPG-induced proliferation (Lawrie et al. Evidence of a role for osteoprotegerin in the pathogenesis of pulmonary arterial hypertension. *Am J Pathol.* 2008;172:256–264) where we defined the chosen dose from a dose-response on proliferation. We have consistently used this for 10+ years. As the reviewer points out, to mitigate the heterogeneity of human cells we used a pooled batch of 3 distinct ‘donors’ of cells, in triplicate (ie n=9). In addition, the validation TaqMan PCR was performed on the same 9, plus 3 new independent mRNA samples from new donors. Given the amount of additional *in vivo* work that was requested, and that we now provide requested data showing that the observed regulation of the OPG target TRAIL by Ky3 is reproduced in rodent lung (Fig 9) we did not do further mRNA expression profiling.

9. *Elevated osteoprotegerin levels have been found to increase apoptosis of endothelial progenitor cells (EPCs) by induction of oxidative stress (Kim JY et al., Arthritis Rheum. 2013). As indicated by the authors (page 5), however, previous studies have implicated circulating mesenchymal progenitor cells such as EPCs in the pathogenesis of PH. Why this paradox? Analysis of the EPC population in these rodents is required to clarify this point.*

The referenced paper (Kim JY et al., Arthritis Rheum. 2013) is the only paper that we can find that describes a pro-apoptotic effect of OPG. In our hands, we have not observed this response on any cell type. We show that OPG induces proliferation and pro-survival signaling. The EPC literature is complex and full of contradictions about what constitutes a circulating endothelial progenitor cell. Most recent studies on EPC have focused (as we have) on the late outgrowth endothelial cells from blood (BOEC) and not early outgrowth or flow cytometry CD34+ isolated cells.

We feel that given the context of the manuscript, and the large body of additional experiments performed that the analysis of the EPC population in these rats is beyond the scope of this current manuscript. We have rather focused in addressing the points raised above concerning the therapeutic efficacy of Ky3.

10. *Tnfrsf11B, the gene encoding osteoprotegerin, is an established HIF target gene (Wu et al., Genes & Dev. 2015). This point has to be taken into account when analyzing the data regarding its upregulation in the sugen+hypoxia and the hypoxic neonatal calf models of PH. In addition, how do the authors explain the fact that Sugden alone increases the level of osteoprotegerin (figure S1)?*

The reviewer correctly highlights that there are reports of OPG being a HIF target. We acknowledge that the data from the SuHx and Calf model will have an upregulation of HIF, however there are multiple reports of HIF being upregulated in samples from patients with PAH. Indeed this is partly why these models are used to investigate parts of the human disease mechanism. This again highlights our approach to using multiple models, with different disease drivers. We highlight the data where the PAH phenotype is not directly induced by hypoxia i.e. HFD-ApoE and monocrotaline. Across all these models the phenotype and response to loss of / blockade of OPG is consistent.

We have not directly investigated the increased OPG in response to Sugen5416 alone, but there are reports of varying response to Sugen5416 amongst rat strains, and evidence that in certain Sprague-Dawley strains Sugen5416 treatment alone is sufficient to induce PAH (Jiang, B. *et al.* Marked Strain-Specific Differences in the SU5416 Rat Model of Severe Pulmonary Arterial Hypertension. *Am. J. Respir. Cell Mol. Biol.* **54**, 461–468 (2016)).

One of the proposed mechanisms by which Sugen5416 induces PAH is thought to be via EC dysfunction triggered by VEGFR2 inhibition. This can then be further exacerbated by hypoxia. It is known from the literature that OPG is co-localised with vWF in the Weibel-Palade bodies of endothelial cells (Zannettino, A. C. W. *et al.* Osteoprotegerin (OPG) is localized to the Weibel-Palade bodies of human vascular endothelial cells and is physically associated with von Willebrand factor. *J. Cell. Physiol.* **204**, 714–723 (2005)). The increase in circulating OPG in response to Sugen 5416 alone is likely due to release from EC, providing further evidence of the credibility of OPG as a drug target for early pathological events leading to PAH.

11. Experiments with Wild-type (wt) littermate mice are missing.

We are not clear to which experiments/figure the reviewer referring here. For the Mouse experiments in Figure 1, A-F utilises a ApoE x OPG cross, where the double knock mice are compared to each individual knock-out (ie littermates). Furthermore there is the inclusion of disease stimulus in each e.g. chow v HFD. In Panels G-O of figure 1 we are directly comparing littermates randomly assigned to receive either HFD, or chow. We have previously reported C57 mice do not develop PAH in response to the HFD used (Lawrie *et al.* Paigen diet-fed apolipoprotein E knockout mice develop severe pulmonary hypertension in an interleukin-1-dependent manner. *Am J Pathol.* 2011;179:1693–1705.). What further controls does the reviewer want? In our mind it would be unethical to include additional C57 mice to all these experiments as it would provide no additional information as to the role of OPG in PAH. In Figure 2 we generate chimeric mice from C57 and OPG^{-/-} littermate mice. We include all the controls including same strain engraftment ie C57 to C57.

MINOR CONCERNS:

1. Statistical signs are missing in Figure 3E.

We apologise for any missing explanation of statistics in the manuscript or figure legends. Figure 3 has changed with this revised version of the manuscript but we have thoroughly checked and corrected any missing statistics.

Reviewer #2 - expert in therapeutic antibodies (Remarks to the Author):

The authors describe a novel therapeutic antibody strategy to neutralize OPG for pulmonary hypertension. The manuscript is technically sound, but requires a few additional details and changes prior to publication.

1. The authors state their antibodies are high affinity. Please provide SPR or other data to support this.

Please see the table in Fig S3e which contains a summary of the relative affinities and true KD values determined by SPR, and response to point 3 in response to Reviewer 1. Since the target antigen is dimeric avidity will be a strong component of the binding kinetics and therefore the values generated using bivalent full IgG antibody molecules are labelled as “relative affinities” and not true KD values. This is also the reason why the off-rates for some interactions could not be determined and KDs could not be calculated (CNROR means “could not resolve off-rate”). True 1:1 binding kinetics and KD values using Fab fragments were also determined by SPR. The KD values for the Fabs are 350pM for human, 410pM for cyno and 3.5 nM for rat OPG. Based on this it is correct to assume that the binding activity in humans would be superior to the activity seen in the rat models and that the dosing required to see efficacy effects in rat models maybe an overestimation. The KD value for human OPG is about 10-fold better than for rat OPG

Fig S3 - Generation and screening for lead candidate human anti-OPG therapeutic monoclonal antibodies. (a) Conventional intraperitoneal injections as well as a rapid immunisation at multiple sites (RIMMS) immunisation regimes were set up using human and rat recombinant OPG protein as immunogen using the KyMouse^{TM25}. (b) A secondary screen identified high affinity antibodies which were then tested for their neutralisation profiles for the interaction of OPG with TRAIL and RANKL. (c) Epitope binning was then performed and the lead panel of antibodies chosen (d) based on the the best affinity antibody identified from each bin. (e) binding criteria for the 4 lead candidate antibodies. CNROR - could not resolve off-rate.

2. Also, what is the affinity for rat OPG? The authors suggest their *in vivo* studies might not reflect the full potential in humans due to incomplete homology. Is the mAb more potent against human OPG?

These data are now included in Fig S3e and included above in response to point 1 above.

3. How were the four mAbs chosen? Do they block OPG binding to TRAIL or RANKL equally well? Please provide additional *in vitro* characterization. Do they block the interaction with Fas?

We apologise for not make this clearer in the original manuscript. The four mAbs were chosen based on their ability to block OPG binding to RANKL or TRAIL which is summarised in Fig S3. Unfortunately at the time of testing their ability to block OPG-TRAIL and OPG-RANKL binding we had not yet identified FAS as the key receptor, so this was not included in the screen. We now included functional data demonstrating that Ky3 inhibits OPG-FAS-induced signalling and phenotype *in vitro* (Fig 7), and signalling *in vivo* (Fig 9).

We have further expanded this explanation in the manuscript (page 8) as below.

Using the KyMouse™ system²⁸ we generated a diverse panel of high affinity anti-human OPG monoclonal antibodies with cross reactivity to rat and cynomolgus monkey, displaying distinct neutralisation profiles and varying ability to block the interaction of OPG with TRAIL and RANKL (Fig. S3). Selected antibodies were chosen to cover a spectrum of partial and full inhibition of OPG-TRAIL and OPG-RANKL binding. OPG-FAS signalling was examined later (Fig 7). Four candidate anti-OPG antibodies (Fig S3e).

4. Related to #1-3, what antibody property best correlates with *in vivo* efficacy?

Based on the SPR data shown in Fig S3 and further summarised below, complete inhibition of OPG-TRAIL and partial inhibition of OPG-RANKL are the characteristics of the most efficacious antibody.

Clone ID	TRAIL Neutralisation	RANKL Neutralisation
6D07 (KY1)	inhibition	No inhibition
8C10 (KY2)	inhibition	inhibition
15F11 (KY3)	Complete inhibition	Partial inhibition
16G05 (KY4)	Partial inhibition	Partial inhibition

However as discussed above the ability of Ky3 to block OPG-Fas signalling was subsequently tested *in vitro* in functional assays. We have not tested the ability of Ky1, Ky2 or Ky4 to block OPG-Fas signalling or phenotype due to limited availability of those antibodies i.e. only Ky3 was scaled up.

5. Please define what IgG4 PE means and why IgG4 was chosen.

Human IgG4 is naturally void of Fc-mediated effector functions. Fc-mediated functions include killing of cells expressing the target. Although OPG is a soluble factor and there is no perceived mode of action that would justify such an active Fc, a disabled isotype was chosen. “PE” refers the human IgG4 Fc region where amino acids P and E at residues S228 and L235 (EU index) are mutated to stabilises the hinge region and remove residual antibody-dependent cell-mediated cytotoxicity. We apologise for not including this information in the methods of the paper. This has now been corrected.

Selected leads were produced in larger quantity in suspension CHO cells and purified as fully human IgG4 PE (human IgG4 Fc region with mutated to amino acids P and E at residues S228 and L235 (EU index) to stabilise the hinge region and remove residual antibody-dependent cell-mediated cytotoxicity) and assessed in *in vitro* and *in vivo* studies (Fig S3).”

Reviewer #3 - expert in osteoprotegerin (Remarks to the Author):

The manuscript entitled “A novel therapeutic antibody targeting osteoprotegerin attenuates severe experimental pulmonary hypertension” addresses OPG as therapeutic target in PAH. The authors report a large amount of data obtained in vitro and in vivo in different rodent models of PAH, overall supporting the potential of a therapeutic strategy based on an anti-OPG antibody.

The results are of great interest since innovative strategies for PAH are needed. Moreover, interesting mechanistic insights have been included such as the role of Fas receptor in mediating the effects of OPG in PAH.

We thank the reviewer for recognising the importance of this work and their enthusiasm for our work.

I have the following comments:

- since OPG has a pro-inflammatory activity on the vasculature, I was wondering if the authors have assessed the effects of Ky3-treatment on the pro-inflammatory status related to PAH in Ky3-treated animals versus controls (e.g. tissue infiltration of inflammatory cells around the plexiform lesions or expression of inflammatory mediators such as IL-6 or modulation of circulating serum pro-inflammatory chemokines/cytokines)?

We thank the reviewer for highlighting the potential role of OPG in regulating inflammation, and in turn modulating the inflammatory component of PAH. There are many pieces of evidence to suggest that inflammation ‘aggravates’ the pathogenesis of PAH. We now

include new data that provide further evidence that OPG can activate inflammatory signalling. Indeed, we identified an NF- κ B profile in cell stimulated with OPG (Fig 3) and now demonstrate that this is modulated by OPG-Fas (Fig 7C).

In addition we performed CD68 immunostaining to examine the number of macrophages. We can confirm that we did indeed see macrophages recruited to the perivascular regions however we did not detect a reduction in Ky3 treated rats. We did however detect differences in the expression of TRAIL and IL-6 in Ky3-treated rats. We include this data in a new Fig 9 (included above in response to Reviewer 1, point 5).

- in the in vivo experiments Ky3-treated animals show increased circulating OPG levels compared to controls. The authors hypothesize that it might be due to retention of Ky3-bounded-OPG in the circulating system rather than allowing access to vessel wall. Are there evidences of a reduced OPG staining in the tissues of Ky3-treated animals compared to controls in the different PAH models?

We thank the reviewer for highlighting this oversight from our original manuscript. As suggested by the reviewer we have performed immunohistochemical analysis of the lung tissue for OPG. We do indeed see an accumulation of OPG in the small remodelled pulmonary arteries (25-30 μ m) in SuHx and SuHx treated with IgG4 that is reduced in groups treated with Ky3. This data is included in Fig 9.

How were the circulating levels of OPG in the experiment showed in Figure 1 where a polyclonal anti-OPG antibody was used? The complete kinetics of the circulating levels of OPG should be shown in all in vivo experiments for both controls and treated animals. I was wondering if the authors have excluded possible cross-reactivity in the assay used to quantify circulating OPG.

We thank the reviewer for highlighting this important point. We did measure the kinetics of circulating OPG in all *in vivo* experiments and have now included this in Figures 6, 8 and 10. We did also perform numerous studies to rule out interference of TRAIL, RANKL, Fas on the detection of OPG. Neither of which had any effect on OPG detection. We are therefore confident to state that the serum levels of OPG represent the total OPG including ‘free’ OPG as well as ‘bound’ OPG.

- since OPG is the decoy receptor for TRAIL and that TRAIL is involved as well in vascular pathophysiology and in endothelial cell biology with conflicting results and implications, do the authors have any information on the modulation of TRAIL due to Ky3 treatment (in terms of soluble TRAIL and/or transmembrane isoform)?

We agree with the reviewer that it is important to consider the effect of TRAIL and OPG together. Indeed, we have identified a novel mechanism for induction of TRAIL expression

via the OPG-Fas axis. We now added new data specifically demonstrating that this also happens *in vivo*. This new data in Fig 9 C&D directly shows the induction of TRAIL in disease (as we have previously demonstrated) and the reduction in TRAIL following the treatment with Ky3.

As minor points, editing issues should be addressed such as:

- indication of the statistical significance in figures and legends

Check

- the generation of chimeric mice is missing in the material and methods section

We apologise for this omission but have now added the follow text to the methods section.

Where stated bone marrow transplantation (BMT) was performed as previously described^{12,48}. Briefly, male mice, 6 to 8 weeks old, received a sub-lethal dose of whole-body irradiation (1100 rads, split into two doses, 4 hours apart). Irradiated recipients then received 3 to 4 million cells isolated from 4 to 6 week old mice, in Hanks' balanced salt solution, by tail-vein injection. Mice were allowed to recover for 6 weeks after bone marrow transfer prior to induction of PAH.

- Figure S3_panel B: there is only one line-graph

We apologies for this, the 2 data lines actually overlay each other and one was masking the other. We have changed the colouring to make this easier to see in the new figure, now Fig S5.

- the sentence describing figure 1D is not clear (see results, page 2, second paragraph)

Apologies if this was not clear - this now reads

There was no statistically significant difference in cardiac index (CI) between HFD-fed *ApoE^{+/-}* and *ApoE^{+/-}/OPG^{-/-}* (Fig. 1D)

- genes nomenclature

We apologise for inconsistency and have corrected this throughout the manuscript.

- the PAAT acronym is not mentioned in full in the manuscript

Again apologies for not including this PAAT is pulmonary artery acceleration time. This has been rectified.

- in Figure 1F the indications of ABEGV and α -SMA are missing

This has been rectified.

Reviewers' comments:

Reviewer #1 (Remarks to the Author):

The authors have answered the majority of the requests of the three reviewers, and consequently the manuscript has been much improved. However, some points still need to be improved and further discussed:

MAJOR CONCERNS

1-The manuscript would be considerably strengthened if the authors could increase the numbers of animals to at least 5 animals per group (especially in Figure 1 and 2).

2-The authors state that anti-OPG treatments can attenuate the development of PH in multiple rodent models (abstract and results and discussion); however, there is a lack of efficacy of the anti-OPG approach on hemodynamics in two experimental models - the chronic hypoxia (Fig S1) and monocrotaline (Fig S5) PAH models. This point should be clarified and more discussed. Is there any link with the Fas receptor expression in these different models? Does therapeutical delivery of Ky3 in combination with sildenafil or bosentan in these specific animal models may have a beneficial effect on hemodynamics compared to sildenafil or bosentan alone?

3-This reviewer also regrets the absence of data showing a positive effect of Fas siRNA on PA-SMC proliferation and NF-kb activation (figure 7). A condition in which PA-SMCs are only treated with Ky3 is missing.

4-What would be the potential effects of this therapy targeting the OPG-Fas signalling on cardiomyocyte, endothelial and fibrocyte functions?

MINOR CONCERNS

1-The HFD-ApoE^{-/-} mouse model is not a well-recognized model of PAH. Thus, I would recommend adding these results (Figure 1) to the Supplementary data.

2-Figure 5I and 5J are confusing and unnecessary. They should be deleted or more focused on the findings obtained in this study.

Reviewer #2 (Remarks to the Author):

The authors have revised the manuscript and improved the manuscript overall. There are a still several changes that should be made prior to acceptance.

1) In Fig. S3, please provide units of KD with replicate measurements. In c, there is a typo "epitope binning on".

2) Please provide methods for the SPR and biochemical neutralization assays. Additionally, the authors are encouraged to show the data for neutralization of TRAIL/RANKL.

Reviewer #4 - replacement for reviewer #3 (Remarks to the Author):

This is an interesting manuscript showing that anti-OPG Ab therapy attenuates PAH by modulating vascular remodelling. The interaction between Fas and OPG is novel, and the involvement of Fas in PAH has not been described.

Overall, the authors have addressed most of the concerns.

Point 1. Changes in inflammation

NfKB activation can have inflammatory and anti-inflammatory roles. Why was a no-treatment control included in the Kinex array assays? It is difficult to interpret this data; the heat map (Fig 3) suggests that OPG increases p50 but reduces p65 expression (60 min v. 10 min). Has this data been validated? How was Nfkb activity assessed in Fig 7? Here, OPG is increasing NFkB activity. It is difficult to compare without knowing the methodological details.

Point 3: Circulating OPG levels

Fig 6 – It is difficult to differentiate between the 2 hatched lines. Could the authors please modify to make easier to distinguish treatments.

Have the authors measured TRAIL, RANKL, Fas, in the circulation from these mice?

Point 4: modulation of TRAIL due to Ky3

Fig 5 suggests that the increase in TRAIL by OPG-Fas complex is a direct effect. Although I feel this requires more discussion. What happens to circulating TRAIL levels (see above)?

Additional comments:

1. It would be worthwhile to have greater discussion on the FAS/OPG interaction – which I feel is the major novel aspect of this work. OPG is generally considered a decoy receptor that antagonizes the natural ligand. What do the authors think is going on here? Is OPG antagonizing FasL's interaction with Fas in this model? Or is OPG acting as a ligand? What happens to FasL expression (qPCR and circulating) in the model and treatment groups? What happens to Fas and FasL mRNA expression in response to OPG from the microarray?
2. please indicate levels of chimerism in BMT?
3. The microarray data needs to be deposited into a public repository (in a private setting until publication).

Thank you for re-reviewing our manuscript and providing thoughtful comments. Below is a point-by-point response to the comments/queries raised with the reviewers' comments in *italics*, our response in plain text and changes in the revised manuscript text in **red** below, and in the revised manuscript.

Reviewers' comments:

Reviewer #1 - expert in PAH (Remarks to the Author):

Reviewer #1 (Remarks to the Author):

The authors have answered the majority of the requests of the three reviewers, and consequently the manuscript has been much improved. However, some points still need to be improved and further discussed:

MAJOR CONCERNS

1-The manuscript would be considerably strengthened if the authors could increase the numbers of animals to at least 5 animals per group (especially in Figure 1 and 2).

We appreciate the concern raised by this reviewer on the number of animals used in some of the experimental groups presented in figures 1 & 2. We attempted to address this in our previous resubmission by highlighting the number of whole experiment repeats in some of those models but we appreciate this only applies to Figure 1. Unfortunately we no longer have access to the OPG^{-/-}, or ApoE^{+/-}/OPG^{-/-} colonies, so we regret that we cannot easily increase the n-number. We also highlight to this reviewer the high quality work by Jia *et al* (Ref 15) where a genetically distinct OPG null mouse exactly phenocopies our data (Fig. S1). These data, from a different lab provide the best possible reproducibility data confirming that loss of OPG is protective in the Mouse SuHx model.

Specifically, and relevant to figure 2, we have recently generated a new OPG floxed mouse to allow further investigation into the bone marrow source of OPG. This mouse is still being crossed with the relevant cre-driver lines prior to characterisation. We feel that this data is very much beyond the scope of this manuscript - for which we would like to keep the focus on the new Ky3 drug developed.

2-The authors state that anti-OPG treatments can attenuate the development of PH in multiple rodent models (abstract and results and discussion); however, there is a lack of efficacy of the anti-OPG approach on hemodynamics in two experimental models - the chronic hypoxia (Fig S1) and monocrotaline (Fig S5) PAH models. This point should be clarified and more discussed. Is there any link with the Fas receptor expression in these different models? Does therapeutic delivery of Ky3 in combination with sildenafil or bosentan in these specific animal models may have a beneficial effect on hemodynamics compared to sildenafil or bosentan alone?

We note the reviewer's comment on the lack of efficacy of anti-OPG treatments in reducing haemodynamic measurements in 2 models (mouse Hx Fig S1E and monocrotaline, Fig S5E). We explain this with reference to the mechanism of action of OPG where alterations in vascular remodelling due to anti-OPG take longer to impact upon haemodynamic changes than vasodilator treatments like sildenafil. We highlight the novelty of our work developing anti-OPG therapy (the Ky3 antibody) targeting a different mechanism of action (MOA) compared to current standard of care (SOC). Specifically, we believe that Ky3 targets the

pulmonary vascular remodelling aspects of the disease and has minimal (if any) direct effect on acute vasoreactivity. As such, haemodynamic changes in these specific models will take much longer to work through and be observed than with a SOC vasodilator therapy. This is the case in the monocrotaline model where we saw a significant reduction in pulmonary vascular remodelling but no significant reduction in pressure. Importantly, the data in figure S1 highlighted by the reviewer are unlikely to be representative of a therapeutic anti-OPG approach, as these experiments were actually performed in whole body knock-out mice which have never expressed OPG and may therefore have some underlying genetic compensation. Furthermore, we do observe downward trends in all PH parameters in the OPG^{-/-} Hx mice compared to wild-type littermate Hx mice. Our focus for this specific experiment (Fig S1) was to assess the phenotype in the SuHx mice, the Hx alone group was included merely as a control and not a primary focus for the experiment. The fact that we see a significant reduction in remodelling and right ventricular pressure in the mouse SuHx experiments but not with hypoxia alone suggests that this approach (anti-OPG) may be more appropriate where the model displays more advanced pulmonary vascular remodelling. It is well known that the mouse Hx model only weakly responds to hypoxia and displays subtle pulmonary vascular muscularisation. With respect to the point identified by the reviewer and the above arguments, we have altered our ‘claim’ in the abstract, results and discussion to make it clearer that we are specifically referring to the effect upon pulmonary vascular remodelling when we state we have efficacy in multiple rodent models. We have edited the text to say

“...anti-OPG treatments can attenuate the development of pulmonary vascular remodelling associated with PAH in multiple rodent models.”

The reviewer also raises the question of whether combination therapy of Ky3 with sildenafil or bosentan would show beneficial effect in either the aforementioned Hx mouse or monocrotaline rat models. As highlighted as part of the response above, our focus for the development of Ky3 has been on anti-pulmonary vascular remodelling, not primarily haemodynamics.

With regard to the monocrotaline (Mct) rat model, we have every confidence that the combination of Ky3 with either sildenafil or bosentan would have a beneficial benefit over each alone since we show benefit for pulmonary vascular remodelling with Ky3 and haemodynamics with sildenafil. This being said, it is important to also note that as reported by Stachar et al (2012 in AJRCCM), there is no correlation between haemodynamics and pulmonary vascular remodelling in the Mct rat model. Indeed, even in human studies, although mPAP is a diagnostic requirement, it is a poor predictor of outcome so the focus solely on demonstrating a haemodynamic response when we have already demonstrated a positive effect on pulmonary vascular remodelling is in our opinion a mute point. Again we argue that we have already demonstrated efficacy in both models with respect to beneficial effects on pulmonary vascular remodelling.

We strongly believe that the mouse Hx model is not suitable for assessing drug efficacy particularly for those therapies targeting pulmonary vascular remodelling. The therapeutic window is too small (pulmonary vascular remodelling very subtle) and existing vasodilator therapies, particularly at the doses given in these models would leave no ‘window’ (particularly haemodynamics) upon which to demonstrate any additional therapeutic benefit.

Finally, the combination therapy experiments are not trivial to perform, would require a 70-80 rat study, and take about 6-9 months to perform at a cost of ~£100,000 including human resource. Since we have already demonstrated a reduction in pulmonary vascular remodelling with Ky3 in 2 rat models and further reduction in RVSP and pulmonary vascular remodelling in combination with PAH therapies in the SuHx model, a further combination study in the Mct model would have very little added scientific benefit.

Finally, the PAH research field has openly suggested we move away from the monocrotaline model, and has most recently requested that all types of preclinical therapeutic studies be performed in the SuHx rat. We strongly feel that a further mouse study is not warranted.

We would also highlight that as well as undertaking a number of batch repeats with Ky3 ourselves, Ky3 has also recently been tested at an independent CRO by Kymab. These commercially sensitive data are available to share confidentially with the Editor (to share with this reviewer in strict confidence, if required) and they demonstrate a beneficial dose response of Ky3 on reducing RVSP and increasing Cardiac Output.

We note the question about Fas expression being different between these two models, and we have explored this. In both models, as in human PAH (Fig 4F) we can demonstrate that Fas is increased in PAH rats compared to non-PAH rats in both models, we have included these new data in Fig 4 (G & H). Since we demonstrate beneficial effects on pulmonary vascular remodelling in both models, and Fas is upregulated at the gene and protein level in both models we have not attempted to compare directly between them. As explained previously in response to this reviewer, we believe that the difference in haemodynamic effect between the models reflects the different treatment durations. Specifically, the shorter treatment duration in the Mct model was not sufficiently long for the effect on pulmonary vascular remodelling to impact upon measured RVSP.

We have added the following text to the manuscript to describe these new data in Figure 4.

Investigation of rat lung isolated from control (saline) and monocrotaline rats, as well as control (normoxic) and SuHx rats also demonstrate a significant increase in expression of both Fas gene expression (Fig. 4G) and protein expression within remodelled pulmonary arterioles (Fig. 4H). We have updated the figure legend accordingly.

3-This reviewer also regrets the absence of data showing a positive effect of Fas siRNA on PA-SMC proliferation and NF-kb activation (figure 7). A condition in which PA-SMCs are only treated with Ky3 is missing.

We understand this request from the reviewer however, as we highlighted in Figure 5H, the blockade of Fas signalling induces the expression of TRAIL, which in turn promotes the proliferation of PASMOC making these data difficult to interpret. We therefore presented data on migration since due to the shorter time course for *in vitro* migration assays (hours vs days for proliferation), TRAIL induction is less of an issue, and it is therefore easier to interpret the data arising. In Figure 5, we demonstrate that a neutralising Fas antibody inhibits OPG-induced migration and proliferation (when combined with an anti-TRAIL antibody). In Figure 7B, we replicate this anti-migration effect with a Fas siRNA but since the focus of this manuscript is the development of an anti-OPG antibody (Ky3) as a therapeutic strategy, we

chose to focus upon demonstrating the effect of the Ky3 antibody on proliferation and on NF- κ B activation to elucidate the mechanism of action further.

The requested control data is provided in Figure 7A. We examine the effect of the IgG4 control and Ky3 separately on SFM media, PDGF and OPG-stimulated cells and observe no effect when cells are treated only with Ky3.

4-What would be the potential effects of this therapy targeting the OPG-Fas signalling on cardiomyocyte, endothelial and fibrocyte functions?

We appreciate the enthusiasm of the reviewer in asking for data on OPG-Fas signalling on cardiomyocyte, endothelial and fibrocyte function but feel that this is another programme of work that would move the focus of the paper away from its as yet unpublished role in pulmonary vascular remodelling. As stated above, we have recently generated a floxed OPG mouse and plan to fully explore the questions raised in the years to come but this is beyond the scope of this already substantial body of work.

MINOR CONCERNS

1-The HFD-ApoE^{-/-} mouse model is not a well-recognized model of PAH. Thus, I would recommend adding these results (Figure 1) to the Supplementary data.

We appreciate the concerns of the reviewer that this model is not widely used but we do point to increasing evidence to suggest of aberrant lipoprotein (ApoE) levels associated with human IPAH (Rhodes, C. J. et al (2017) Plasma proteome analysis in patients with pulmonary arterial hypertension: an observational cohort study. *The Lancet Respiratory Medicine*, 5(9), 717–726.), increase in prevalence of insulin resistance (Zamanian, R. T., et al (2009). Insulin resistance in pulmonary arterial hypertension. *The European Respiratory Journal*, 33(2), 318–324) and the changing demographics towards and increase in an older population of patients with IPAH (Ling, Y. et al (2012). Changing demographics, epidemiology, and survival of incident pulmonary arterial hypertension: results from the pulmonary hypertension registry of the United Kingdom and Ireland. *American Journal of Respiratory and Critical Care Medicine*, 186(8), 790–796).

There have also been a number of articles using this model published by ourselves (Hameed, A. G. et al (2012) Inhibition of tumor necrosis factor-related apoptosis-inducing ligand (TRAIL) reverses experimental pulmonary hypertension. *The Journal of Experimental Medicine*, 209(11), 1919–1935 and Lawrie, A. et al (2011) Paigen diet-fed apolipoprotein E knockout mice develop severe pulmonary hypertension in an interleukin-1-dependent manner. *The American Journal of Pathology*, 179(4), 1693–1705), and initially by the Stanford group (Hansmann, G. (2007). Pulmonary arterial hypertension is linked to insulin resistance and reversed by peroxisome proliferator-activated receptor- γ activation. *Circulation*, 115(10), 1275–1284). Furthermore, the Vanderbilt group have published data implicating that the addition of western diet to BMPR2 heterozygous mice results in a more severe PAH phenotype (Talati, M. H. et al (2016). Mechanisms of Lipid Accumulation in the Bone Morphogenetic Protein Receptor Type 2 Mutant Right Ventricle. *American Journal of Respiratory and Critical Care Medicine*, 194(6), 719–728.). This model is therefore in our

view no less relevant to human disease than all the other models used throughout this study. More importantly, this is not a self-resolving mouse model and the data presented in Figure 1 provided the very first proof of concept that targeting OPG with an anti-OPG strategy was a tractable therapeutic approach. We therefore argue that since this is raised as a minor concern only that these key data remain within the main manuscript.

2-Figure 5I and 5J are confusing and unnecessary. They should be deleted or more focused on the findings obtained in this study.

We apologise if the reviewer finds these diagrams confusing. They were an attempt to summarise the novel OPG-Fas interaction, the direct signaling from Fas to TRAIL, and highlight the concept of targeting OPG. We have removed them from this revised version and would be happy to take advice from the editor on redrawing a summary schema upon acceptance, if that was felt to benefit the manuscript.

Reviewer #2 (Remarks to the Author):

The authors have revised the manuscript and improved the manuscript overall. There are a still several changes that should be made prior to acceptance.

1) In Fig. S3, please provide units of KD with replicate measurements. In c, there is a typo "epitope binning on".

We thank the reviewer for their input and for spotting the typo on Fig S3c - this has now been corrected in this revised manuscript. The units for KD are nM (nanomolar). Unfortunately, the SPR experiment was only performed once on purified chimeric hybridoma antibodies. We have added the following text to the figure legend of Fig S3 to hopefully make this clearer.

binding criteria for the 4 lead antibodies as identified in a single experiment using purified chimeric hybridoma antibodies. nM = nanomolar, CNROR = could not resolve off-rate.

2) Please provide methods for the SPR and biochemical neutralization assays. Additionally, the authors are encouraged to show the data for neutralization of TRAIL/RANKL.

We thank the reviewer for their comments on the binding assay. This assay was designed to determine whether the anti-OPG antibody had any impact on the binding of OPG with RANKL or TRAIL. The assay simply asks the question; does binding of an anti-OPG antibody to OPG impact the binding of OPG to RANKL, or TRAIL? The answer varies depending on the epitope of the anti-OPG antibody used, and was assessed by a competition Homogeneous Time Resolved Fluorescence (HTRF) assay. We have now added the methods for the HTRF assay, and SPR to the Supplemental Methods, and include below, and apologise for their omission from the prior submission.

Homogeneous Time Resolved Fluorescence (HTRF) assay

The OPG/TRAIL HTRF assay was prepared using a starting concentration of 120 nM of each antibody sequentially diluted 1:3 to generate a 11-point curve (0.002 - 120 nM). 5 μ L of each

dilution was transferred to a white 384 well, low-volume, non-binding surface polystyrene plate (Greiner) and mixed with 5 μL of 4 nM human OPG-Fc prepared in HTRF assay buffer (PBS (Sigma) + 0.53 M KF (Sigma) + 0.1% w/v BSA (Sigma)). 5 μL of TRAIL diluted to 16 nM in HTRF assay buffer were then added to assay plates with the exception of control wells used to detect non-specific binding. Before adding the detection solution, plates were incubated for 30 minutes at room temperature to allow receptor-ligand interaction to occur. 10 μL of a solution composed of anti-human Fc D2 (Cisbio) and anti-Flag cryptate (Cisbio) at 1/100 in HTRF assay buffer was added to 5 assay plates.

The HTRF assay to identify antibodies able to inhibit OPG/RANKL interaction was set up following the same method but using 20 nM hOPG-Fc (Seq ID No:154) mixed with 55.6 nM 647-RANKL. Detection solution was prepared using anti-Fc cryptate (Cisbio) diluted at 1/100 in HTRF assay buffer. Control curves were set up using anti-OPG antibody MAB805/Mouse IgG1 prepared at 120 nM working concentration in HMM and diluted 1:3, 11 dilution points (0.002 - 120 nM). Plates were incubated for 1 hour at room temperature and protected from light before measurement of wells at 620 nm and 665 nm emission wavelengths using an EnVision plate reader (Perkin Elmer). Data were analysed by calculating 665/620 ratio (Equation 1) and % Specific Binding (Equation 3) for each sample.

Equation 1: Calculation of 665/620 ratio = (sample 665/620 nm value)

Equation 3: Percentage of % TRAIL and RANKL specific binding Using 665/620 nm ratio (see equation 1) (HTRF) % of specific binding = $\frac{\text{sample value} - \text{non-specific binding}}{\text{total binding} - \text{non-specific binding}} \times 100$

Surface plasmon resonance (SPR)

Affinity of purified anti-OPG antibodies were assessed by label-free surface plasmon resonance (SPR). This analysis was carried out on the ProteOn XPR36 (BioRad) array SPR machine. An anti-mouse IgG capture surface was created on a GLC biosensor 25 chip using amine coupling of an anti-mouse IgG (GE Healthcare). Test antibodies were captured on this surface and human, rat and cyno OPG (produced in-house, Kymab) were used as analyte. The assay was carried out at 25°C using HBS-EP (Teknova H8022). Buffer alone was used to reference the binding sensorgrams. The data was analysed using the 1:1 model inherent to the ProteOn XPR36 analysis software. All the affinity determinations were performed with purified hybridoma material so (human mouse chimera).

Reviewer #4 - replacement for reviewer #3 (Remarks to the Author):

This is an interesting manuscript showing that anti-OPG Ab therapy attenuates PAH by modulating vascular remodelling. The interaction between Fas and OPG is novel, and the involvement of Fas in PAH has not been described.

Overall, the authors have addressed most of the concerns.

Point 1. Changes in inflammation

NfKB activation can have inflammatory and anti-inflammatory roles. Why was a no-treatment control included in the Kinex array assays? It is difficult to interpret this data; the heat map (Fig 3) suggests that OPG increases p50 but reduces p65 expression (60 min v.

10 min). Has this data been validated? How was Nfkb activity assessed in Fig 7? Here, OPG is increasing NFkB activity. It is difficult to compare without knowing the methodological details.

We thank the reviewer for the succinct summary of our work highlighting the key novel data. We also acknowledge the limitation in large screening assays aimed at elucidating signaling mechanisms e.g. the kinex array (Fig S2 and Fig 3). As described, these data were performed using primary cells that are by nature heterogenous in their response. We agree that interpretation of these data needs careful consideration and validation.

We apologise for not more explicitly explaining the heat maps in the figure legend. Like standard microarray heat maps these display relative expression to an unstimulated control, samples shown as black represents no change, blue is increased expression and yellow is a decrease in expression relative to un-stimulated cells at the same time point. The no treatment data are not shown as they would all simply be black. We have added the following text to the figure legend to make this clearer.....**at 10 and 60 minutes expressed as a ratio to unstimulated controls from the same time point.**

Our view is that this screen provides a first clue into the pathway (e.g. cell cycle, NF-kB) but that this must be followed up using alternative methodology to further elucidate the mechanism. Indeed, this is why we performed a number of experiments to validate the arrays (Fig S2, Fig 3 D&E) using traditional western immunoblotting (Fig 3F) and an NF-kB activation assay (Fig 7C). We apologise for not including the details of the NF-kB activation assay within the methods of the manuscript. This was an oversight on our part and we have now included the details below, and in the revised methods section of the manuscript.

NF-kB Activation Assay

PASMCs were seeded into 96 well plates (0.5×10^4 cells/well) and allowed to adhere for 24 hours (37°C, 5% CO₂). Cells were then transfected with 100 ng/well inducible NFkB responsive firefly luciferase reporter and constitutively active Renilla construct mixture using the Cignal reporter assay kit (Qiagen) and Lipofectamine 2000 transfection reagent (Invitrogen) and incubated for 24 hours (37°C, 5% CO₂). Media was then renewed in the presence or absence of stimulation with OPG (30 ng/ml, R&D Systems) with or without 1500 ng/ml of Ky3 or control IgG4 antibodies. Luciferase activity was detected following 48 hours stimulation using Dual-Glo luciferase assay system (Promega).

Our data indicate that OPG signalling via Fas induces a pro-inflammatory response. We are currently exploring further the canonical and non-canonical NF-kB individual protein components activated by OPG to identify the key transcriptionally activated genes. This work is at an early phase and is currently beyond the scope of this manuscript.

Point 3: Circulating OPG levels

Fig 6 – It is difficult to differentiate between the 2 hatched lines. Could the authors please modify to make easier to distinguish treatments.

We apologise that it was not easier to distinguish between groups in this figure, we have altered the colour to make this difference more easy to identify.

Have the authors measured TRAIL, RANKL, Fas, in the circulation from these mice?

Unfortunately, these assays are more complex to perform than it might first seem. Firstly, while there are good assays available for mouse and human TRAIL and RANKL, we have found available rat assays (these are rat models) to be unreliable at measuring levels in plasma/serum. Secondly, the protein-protein complexes formed between OPG-TRAIL and OPG-RANKL can make it tricky to interpret these assays, particularly when they are not well characterised for interference.

Point 4: modulation of TRAIL due to Ky3

Fig 5 suggests that the increase in TRAIL by OPG-Fas complex is a direct effect. Although I feel this requires more discussion. What happens to circulating TRAIL levels (see above)?

We agree that the data presented in Fig 5E indicates the OPG binding to Fas causes an increase in TRAIL expression, and that this over time results in increase TRAIL protein expression, sufficient to induce proliferation (Fig 5H). The relationship between cell expressed and circulating TRAIL is complex. TRAIL is composed of 281 amino acids and has characteristics of a type II transmembrane protein, with the C-terminal extracellular domain proteolytically cleaved from the cell surface. TRAIL then forms a homotrimer that can interact with its receptors (inc OPG). The involvement of proteolytic cleavage to generate circulating TRAIL complicates the relationship between expressed and circulating TRAIL levels. We previously demonstrated that tissue-derived TRAIL is required for mice to develop PAH (Hameed, A. G. et al (2012) *The Journal of Experimental Medicine*, 209(11), 1919–1935.) but as TRAIL is also expressed by immune cells the relevance of circulating versus tissue expressed TRAIL is unclear.

We have added the following text to the discussion of the manuscript.

The implication that OPG can regulate the local expression of TRAIL within the vessel wall fits with our reports demonstrating that TRAIL³⁹, and specifically tissue-derived TRAIL¹² is required for mice to develop PAH. Of note, TRAIL was also recently described to be an important member of an immune cluster of circulating proteins that defined poor prognosis in patients with mixed aetiology PAH⁴⁵. The relationship between cell expressed, and circulating TRAIL is however complex. TRAIL is widely expressed, including by immune cells and circulating ‘soluble’ TRAIL requires proteolytic cleavage of the C-terminal extracellular domain of the transmembrane TRAIL protein. Whether disease is mediated by locally expressed and ‘retained’ TRAIL or by released circulating TRAIL remains unclear.

Additional comments:

1. It would be worthwhile to have greater discussion on the FAS/OPG interaction – which I feel is the major novel aspect of this work. OPG is generally considered a decoy receptor that antagonizes the natural ligand. What do the authors think is going on here? Is OPG antagonizing FasL’s interaction with Fas in this model? Or is OPG acting as a ligand? What happens to FasL expression (qPCR and circulating) in the model and treatment groups? What happens to Fas and FasL mRNA expression in response to OPG from the microarray?

We agree with the reviewer that the Fas-OPG interaction is the major novel finding from this study, although the development of our novel antibody should not be overlooked. It was our feeling all along that based on the phenotype response in cells to OPG, it was working as a ligand. This is explicitly why we performed the membrane binding protein screen. To make

this clearer we have added the following sentence to the results section on Page 6 discussing this screen.

Given the effect of OPG on cell phenotype, and the intracellular signalling identified, we felt that OPG may be acting as a ligand and signaling through a previously undescribed receptor.

We do believe that OPG binding to Fas can antagonise FasL-Fas binding, indeed in Figure 5F we demonstrate that in a FasL induced apoptosis assay OPG can antagonise FasL induced apoptosis. We have added some text to the description of this to make this clearer.

“Pre-incubation of HT1080 cells with OPG significantly blocked both TRAIL but also FasL-induced apoptosis, as measured by Caspase3/7 activation (Fig. 5F) indicating that OPG can antagonise FasL-Fas binding”

There was no significant difference (FDR corrected p-value) in either Fas or FasL gene expression in OPG-stimulated PSMCs from the arrays. However we do highlight that independently Fas expression is increased in Human and rodent PAH cells and tissues (Fig 4)

2. please indicate levels of chimerism in BMT?

We have significant experience of BMT experiments and our average percentage conversion following bone marrow transplantation based on chromosome painting is 95 +/- 2.6%. We did not determine the chimerism in these specific mice but given the clear phenotypic differences, and differences in OPG expression shown in Fig 2 we are confident of a similar level of chimerism to our previously published work (Hameed, A. G., Arnold, N. D., Chamberlain, J., Pickworth, J. A., Paiva, C., Dawson, S., ... Lawrie, A. (2012). Inhibition of tumor necrosis factor-related apoptosis-inducing ligand (TRAIL) reverses experimental pulmonary hypertension. *The Journal of Experimental Medicine*, 209(11), 1919–1935).

3. The microarray data needs to be deposited into a public repository (in a private setting until publication).

We agree and are in the process of uploading the microarray data onto GEOarchive and will make this public once the paper is published. Once we have the reference we will update this in the data availability statement.

REVIEWERS' COMMENTS:

Reviewer #1 (Remarks to the Author):

-

Reviewer #2 (Remarks to the Author):

The paper is now suitable for publication.

Reviewer #4 (Remarks to the Author):

The authors have addressed all my concerns. The work is ready for publication. Congratulations.

Mary Kavurma

REVIEWERS' COMMENTS:

Reviewer #1 (Remarks to the Author):

-

Reviewer #2 (Remarks to the Author):

The paper is now suitable for publication.

Reviewer #4 (Remarks to the Author):

The authors have addressed all my concerns. The work is ready for publication. Congratulations.

Mary Kavurma

We are pleased to hear that the reviewers are satisfied with our previous revisions and response.